# Nonporous amorphous superadsorbents for highly effective and selective adsorption of iodine in water

Wei Zhou[1], Aimin Li[1], Min Zhou[1,2], Yiyao Xu[1], Yi Zhang[1] & Qing He ⬤[1] ✉

Adsorbents widely utilized for environmental remediation, water purification, and gas storage have been usually reported to be either porous or crystalline materials. In this contribution, we report the synthesis of two covalent organic superphane cages, that are utilized as the nonporous amorphous super-adsorbents for aqueous iodine adsorption with the record–breaking iodine adsorption capability and selectivity. In the static adsorption system, the cages exhibit iodine uptake capacity of up to $8.41\,g\,g^{-1}$ in $I_2$ aqueous solution and $9.01\,g\,g^{-1}$ in $I_3^-$ ($KI/I_2$) aqueous solution, respectively, even in the presence of a large excess of competing anions. In the dynamic flow-through experiment, the aqueous iodine adsorption capability for $I_2$ and $I_3^-$ can reach up to 3.59 and $5.79\,g\,g^{-1}$, respectively. Moreover, these two superphane cages are able to remove trace iodine in aqueous media from ppm level (5.0 ppm) down to ppb level concentration (as low as 11 ppb). Based on a binding–induced adsorption mechanism, such nonporous amorphous molecular materials prove superior to all existing porous adsorbents. This study can open up a new avenue for development of state–of–the–art adsorption materials for practical uses with conceptionally new nonporous amorphous superadsorbents (NAS).

Radioactive nuclides are a double-edged sword because when they provide us with clean nuclear energy without green-house gas emissions, they produce problematic nuclear wastes that pose a great threat to the environment and humans[1–3]. As the products of plutonium-239 and uranium-235 fission in the nuclear plant, $^{129}I$ and $^{131}I$ are considered as the two most harmful nuclides since the former has ultra-long half-life ($\sim$$1.6 \times 10^7$ years), toxicity, and high mobility in most geological environments while the latter, notwithstanding a shorter half-life span ($\sim$8 days), interferes with human metabolic processes due to strong radiation[4,5]. As such, of particular challenge is the administration of nuclear wastes. Any sudden nuclear accidents, e.g., the Chernobyl (in 1986) and Fukushima (in 2011) nuclear disasters, could lead to the release of large quantities of radioactive iodine, including $^{129}I$ and $^{131}I$, into the atmosphere as gaseous $I_2$ and water bodies as $I_2$ or $I_3^-$, posing serious threat to people's safety and health[6]. Thus, high-performance materials for iodine capture and remediation from, inter

alia, aqueous effluents with fast kinetics and high adsorption capacity are urgently needed.

Over the past decade, a myriad of materials have been established for iodine adsorption[7–12]. Most of such adsorption materials, more often than not, are reported to be porous sorbents including porous amorphous materials and porous crystalline materials (Fig. 1a, b)[7,8,11]. For instance, porous activated carbon and zeolites alike are the most commonly used adsorbents for trapping iodine because of their low cost, easy availability, and strong affinity[7,13–15]. Nevertheless, such conventional adsorbents are still far from ideal due in large part to their unsatisfying performance in iodine capture (e.g., relatively low iodine loading capacity), inter alia, in aqueous solutions. Excitingly, recent years have witnessed the emergence and dynamic growth of new iodine adsorption materials, such as porous organic polymers (POPs)[16–20], metal-organic frameworks (MOFs)[8,12,21–24], covalent organic frameworks (COFs)[25–31], and porous organic cages (POCs)[32–36], with

[1]State Key Laboratory of Chemo/Biosensing and Chemometrics, College of Chemistry and Chemical Engineering, Hunan University, Changsha 410082, P. R. China. [2]College of Chemistry and Chemical Engineering, Hunan Normal University, Changsha 410081, P. R. China. ✉e-mail: heqing85@hnu.edu.cn

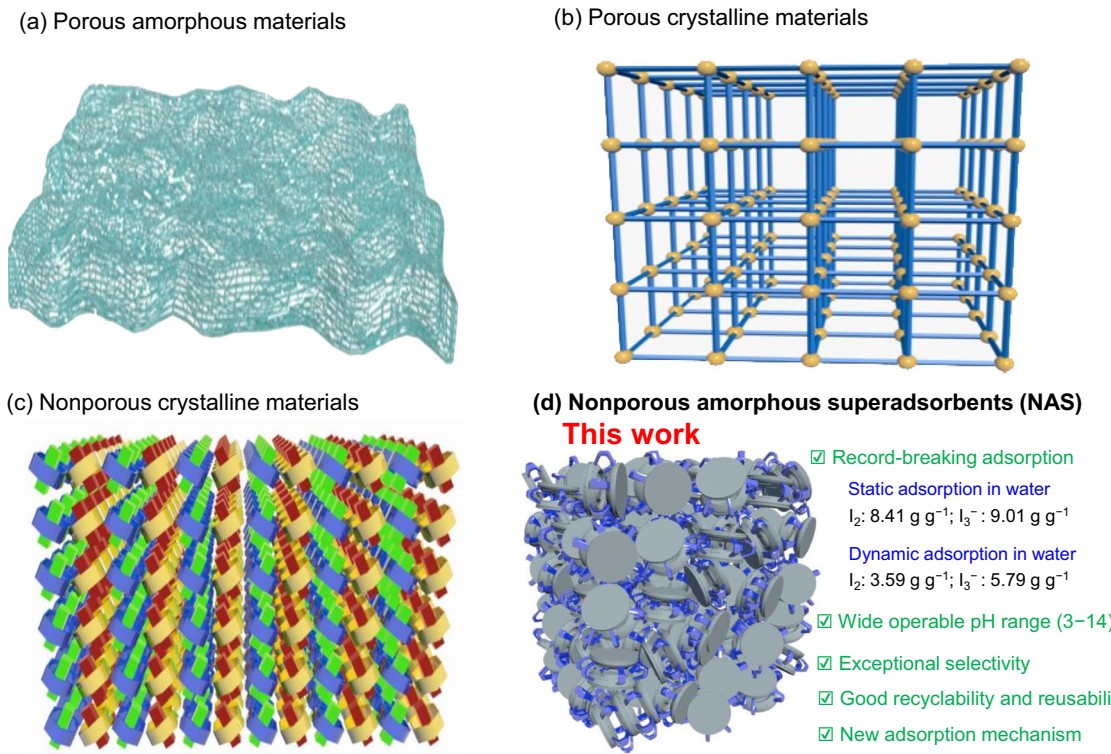

(a) Porous amorphous materials

(b) Porous crystalline materials

(c) Nonporous crystalline materials

**(d) Nonporous amorphous superadsorbents (NAS)**
**This work**

☑ Record-breaking adsorption
Static adsorption in water
$I_2$: 8.41 g g$^{-1}$; $I_3^-$ : 9.01 g g$^{-1}$
Dynamic adsorption in water
$I_2$: 3.59 g g$^{-1}$; $I_3^-$ : 5.79 g g$^{-1}$
☑ Wide operable pH range (3−14)
☑ Exceptional selectivity
☑ Good recyclability and reusability
☑ New adsorption mechanism

**Fig. 1 | Four types of adsorption materials.** Classical adsorption materials: **a** porous amorphous materials; **b** porous crystalline materials; **c** Nonporous crystalline materials and **d** new superphane-based nonporous amorphous superadsorbent (NAS) materials reported in this work.

decent to excellent iodine uptake capability. However, most of those reported materials mainly showcase excellent performance in gaseous iodine, e.g., $I_2$, but unsatisfactory iodine uptake capability (<1.0 g g$^{-1}$) in aqueous media largely due to the interference of water molecules with a noticeable exception[7–9,11,12,37]. Only a few of them were found able to capture iodine from water with high iodine uptake capacity[17,22,38–40]. For example, in 2022, Sessler and coworkers reported a series of porous polymer networks crosslinked with calix[4]pyrrole scaffolds for effectively adsorbing iodine with the uptake capacity of 3.24 g g$^{-1}$ and fast capture kinetics from a concentrated aqueous KI/$I_2$ solution[19]. Simultaneously, Chattopadhyay et al. described porous poly-aminoamide systems for multimedia iodine adsorption with the static iodine adsorption capacity of 5.59 g g$^{-1}$ in aqueous medium[41]. Ma and coworkers established a variety of undoped covalent organic framework aerogels for enhanced iodine uptake, wherein the highest static iodine-adsorbing value was claimed to be ca. 7.13 g g$^{-1}$ in KI/$I_2$ aqueous solution[29]. Impressive are these recent advances in terms of iodine adsorption capability, irrespective of the coexistence of interfering species (e.g. competing anions) and the iodine uptake kinetics. Nevertheless, the aqueous iodine adsorption in real-world scenarios, e.g., wastestreams of the nuclear power industry and contaminated seawater by nuclear leakage in sudden nuclear accidents, is much more complicated than initially thought given the troublesome variables, such as a huge excess of various competing anions and extreme pHs. Such harsh operational conditions preclude most of known iodine-adsorbing materials. As such, conceptionally new material systems aimed at practical applications in aqueous iodine uptake in a wide pH range with superior efficacy, exceptional selectivity, and fast adsorption kinetics are urgently needed.

Unlike porous adsorption materials, nonporous solids had long been considered useless in adsorption of species of interest. Until recently, a new class of synthetic macrocycle-based nonporous materials, that is nonporous adaptive crystals (NACs) (Fig. 1c)[42–44], have been reported to be used for uptake of guest species, e.g., iodine[45,46].

Nonetheless, compared with porous adsorption materials, NACs for iodine adsorption reported so far in the literature usually require additional treatment procedures for preparing crystals before use and exhibit less-than-ideal iodine uptake capability of <1.0 g g$^{-1}$, inter alia, in aqueous media[47]. It is worth noting that the discrete as-prepared macrocycles or cages alone in the form of nonporous amorphous solids usually display even poor or negligible guest adsorption capability, as opposed to their crystalline forms. It thus appears that, as usual, nonporous materials, especially in amorphous state, are deemed far inferior to porous sorbents in terms of guest uptake performance. Herein, we report our serendipitous discovery that the game changers, a new class of superphane-based nonporous amorphous molecular materials as superadsorbents (Fig. 1d), were found capable of adsorbing trace iodine from aqueous solutions with exceptionally high efficacy and selectivity. Firstly, a purely covalent organic superphane cage (**SUPE−py−Imine−Cage**) containing imine bonds was synthesized via dynamic self-assembly of a key hexakis-formylpyridine-amide precursor and hexakis(aminomethyl)benzene. **SUPE−py−Imine−Cage** was then found able to be reduced to the corresponding secondary-amine linked superphane cage (**SUPE−py−Amine−Cage**) via a NaBH$_4$ reduction method. Secondly, both superphane cages were ascertained to be nonporous amorphous molecular solids, that were found effective for iodine uptake from both gaseous streams and, in particular, aqueous media even in the presence of various excessive (100-fold) competing anionic species, viz. Cl$^-$, Br$^-$, NO$_3^-$, and SO$_4^{2-}$. Of particular note is that, regardless of the essentially negligible porosity, **SUPE−py−Imine−Cage** exhibited a record-breaking aqueous iodine uptake capability of 9.01 g g$^{-1}$ under static condition. More importantly, both **SUPE−py−Imine−Cage** and **SUPE−py−Amine−Cage** were observed to near-completely eliminate either $I_2$ or $I_3^-$ from aqueous ppm-level iodine (5 ppm) sources down to ppb level (as low as 11 ppb). In the case of the dynamic adsorption, **SUPE−py−Imine−Cage** was able to capture $I_2$ and $I_3^-$ with the iodine uptake capability of 3.59 g g$^{-1}$ and 5.79 g g$^{-1}$, respectively, from aqueous solutions containing excessive

**Fig. 2 | Synthetic route to superphane-based cages SUPE–py–Imine–Cage (7) and SUPE–py–Amine–Cage (8).** The one-pot [1 + 1] imine condensation of key precursor **6** and hexakis-amine **4** yielded **SUPE–py–Imine–Cage** efficiently, which was reduced with NaBH₄ into its secondary-amine version, **SUPE–py–Amine–Cage**.

competing anions, which represents the highest dynamic uptake capacity among all reported adsorbents, to the best of our knowledge. Finally, both organic NAS materials were capable of being recycled and reused for at least five cycles without occurrence of the cage decomposition, indicating the potential practical applications in complete remediation of trace iodine from aqueous pollutants.

## Results and discussion

### Synthesis, structure, and characterization

Superphanes refer to a specific class of molecular constructs in which the two face-to-face benzene rings are connected by six bridges[48]. Until recently, superphanes had featured challenging synthesis, poor availability, and inability to be utilized as supramolecular hosts or adsorbents due to lack of internal voids[49–51]. Over the past three years, our group have successfully established versatile synthetic approaches to preparation of functionalized superphanes, which can used as an emerging class of supramolecular receptors for neutral and anionic guest species alike[52–55]. In continuance with our endeavors to further push superphane chemistry forward, we here designed and synthesized two superphane cages, viz. **SUPE–py–Imine–Cage** and **SUPE–py–Amine–Cage**, furnished with eighteen nitrogen sites uniformly surrounding the three-dimensional cavity. Notably, a bulky and aliphatic tetrabutyl group (tBu) was introduced onto each pyridinyl unit to improve the solubility of the key intermediates and desired superphane cages. We initialized the synthesis with the starting material **1** readily prepared according to the reported literatures (Fig. 2 and Supplementary Figs. 1–9)[56,57]. The formyl group of **1** was protected as its dimethyl acetal by the treatment of **1** with MeOH in the presence of a catalytic amount of *p*-toluenesulfonic acid (PTSA) to form **2**, whose methyl ester group was then hydrolyzed with NaOH to the corresponding carboxylic acid (**3**). Subsequently, direct amide condensation of carboxylic acid **3** and hexakis(aminomethyl)benzene **4** in the presence of 1-ethyl-3-(3-dimethylaminopropyl)-carbodiimide (EDCI),

hydroxybenzotriazole (HOBT) and N, N-diisopropyethylamine (DIPEA) in dry DMF gave a key hexasubstituted benzene intermediate **5** (in yield of 45%), whose six dimethyl acetal groups were deprotected with CF₃COOH in dichloromethane to generate the corresponding hexakis-aldehyde precursor **6** in an excellent yield (92%). Satisfyingly, direct imine condensation of hexakis–aldehyde **6** and hexakis-amine **4** led to the formation of desired imine-bearing superphane cage **7** (**SUPE–py–Imine–Cage**), which was further reduced with NaBH₄ to provide amine-containing superphane cage **8** (**SUPE–py–Amine–Cage**). Both superphane cages (**7** and **8**) were fully characterized by standard spectroscopic means (Supplementary Figs. 1–9).

### Iodine vapor adsorption

Structurally, **SUPE–py–Imine–Cage** has six amide units, six pyridinyl moieties, and six imine bonds while **SUPE–py–Amine–Cage** possesses six amide units, six pyridinyl moieties and six secondary-amine fragments, surrounding the interior cavity. These N-rich features allowed us to postulate that both cages could have high affinity towards I₂ and they could be utilized for I₂ adsorption. To test our hypothesis, the as-prepared **SUPE–py–Imine–Cage** and **SUPE–py–Amine–Cage** were exposed to iodine vapor at 348 K (based on the typical nuclear fuel reprocessing condition of nuclear industry)[58] and a deepening in color was then observed from pale yellow to dark black (top row for the former and bottom row for the latter) within 36 h (Fig. 3a). Quantitatively, the iodine mass uptake increased as a function of time, reaching the iodine saturation by ca. 14 h for **SUPE–py–Imine–Cage** (kinetics (80%): 0.99 g g⁻¹ h⁻¹) and ca. 7 h for **SUPE–py–Amine–Cage** (kinetics (80%): 0.91 g g⁻¹ h⁻¹), respectively (Fig. 3b and Supplementary Fig. 10). These observations led us to conclude that the I₂ vapor adsorption rate of these two cages to reach full capacity (6.02 g g⁻¹ (36 I₂ per cage) for **SUPE–py–Imine–Cage** and 4.63 g g⁻¹ (28 I₂ per cage) for **SUPE–py–Amine–Cage**) is greater than those of other iodine

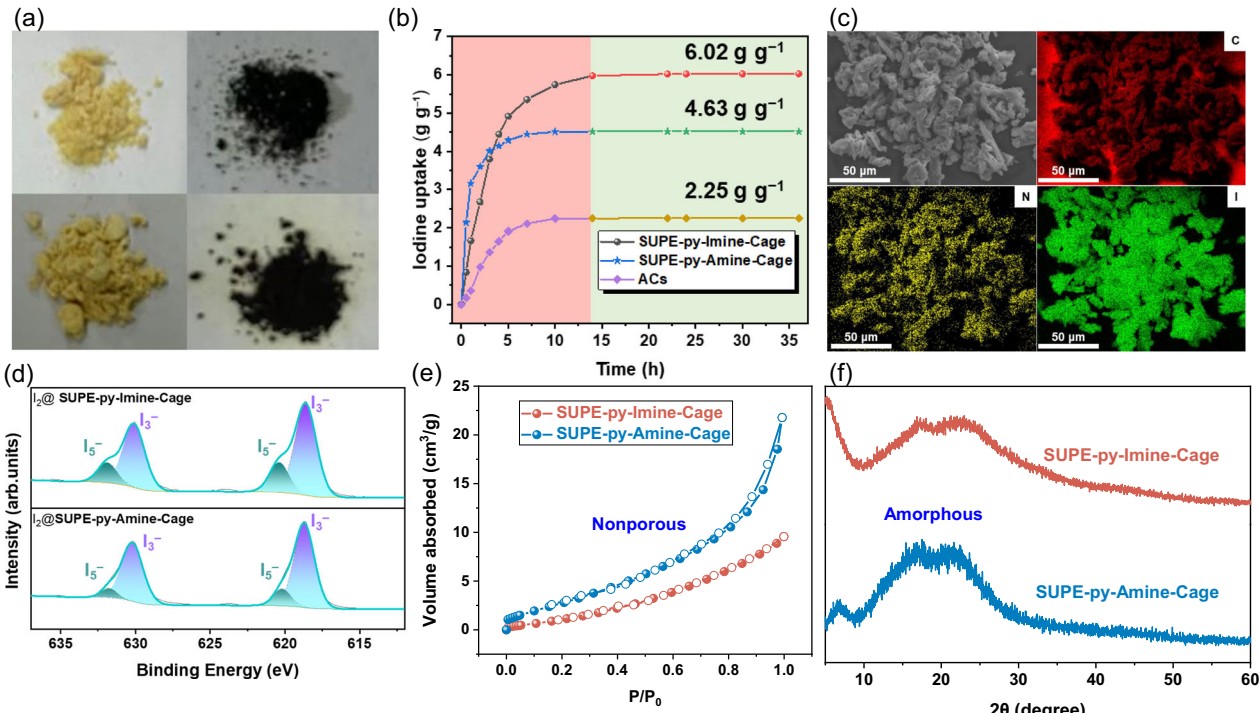

**Fig. 3 | Characterization of iodine vapor adsorption and the nature of adsorbents. a** Photographs showing the color change of **SUPE–py–Imine–Cage** (top row) and **SUPE–py–Amine–Cage** (bottom row) upon exposure to iodine vapor. **b** Time-dependent iodine uptake plots for as-prepared **SUPE–py–Imine–Cage**, **SUPE–py–Amine–Cage** and ACs at 348 K. **c** SEM image and EDS mapping of iodine sorbed **SUPE–py–Amine–Cage**. **d** XPS spectra of I *3d* for I$_2$@**SUPE–py–Imine–Cage** and I$_2$@**SUPE–py–Amine–Cage** derived signals. **e** N$_2$ adsorption (solid symbols)/desorption (open symbols) isotherms at 77 K. **f** Powder X-ray diffraction patterns of **SUPE–py–Imine–Cage** and **SUPE–py–Amine–Cage** solids. Error bars represent SD. $n$ = 3 independent experiments.

adsorbents, e.g., commercially available activated carbons (ACs) with kinetics (80%) of 0.33 g g$^{-1}$ h$^{-1}$ and uptake capability of 2.25 g g$^{-1}$[11,35,47].

We next carried out scanning electron microscopy (SEM), energy-dispersive X-ray spectroscopy (EDS) mapping, Fourier transform infrared (FTIR) spectroscopy, X-ray photoelectron spectroscopy (XPS), and $^1$H NMR spectroscopic analyses to understand in greater details the determinants of iodine sorption. Specifically, the SEM images revealed that both superphane cages are amorphous particles with irregular shape and size (Supplementary Fig. 11). After iodine adsorption, the morphology of the irregular particles for **SUPE–py–Imine–Cage** and **SUPE–py–Amine–Cage** remained essentially unchanged but much fluffier (Fig. 3c and Supplementary Fig. 11). The homogeneous iodine distribution in the cage solids was further ascertained by EDS mapping (Fig. 3c and Supplementary Figs. 12–14). Detailed iodine sorption mechanism was initially evaluated by FTIR spectroscopic analysis. IR spectrum of I$_2$@**SUPE–py–Imine–Cage** revealed that the characteristic peaks at ~1675, 1601 (and 1527) cm$^{-1}$ attributed to the C=O and C=N vibration modes of the carbonyl and pyridinyl units shifted to ~1652, 1597 (and 1525), respectively (Supplementary Fig. 15a). Likewise, similar vibrational frequency shifts were also seen in the case of iodine adsorption utilizing **SUPE–py–Amine–Cage** (Supplementary Fig. 15b). More importantly, as inferred from the XPS spectra of **SUPE–py–Imine–Cage** and **SUPE–py–Amine–Cage** after iodine adsorption, two groups of I *3d* signals are observed (Fig. 3d). In the spectrum of I$_2$@**SUPE–py–Imine–Cage**, signals ascribable to I$_3^-$ were observed at 630.08 and 618.60 eV, respectively. Meanwhile, signals at 631.87 and 620.34 eV could reflect the occurrence of I$_5^-$ species. In contrast, both cages alone showcased a single XPS peaks of *N 1s* at 398.71 and 398.99 eV (C–N bond of the pyridinyl) for **SUPE–py–Imine–Cage** and **SUPE–py–Amine–Cage**, respectively (Supplementary Fig. 16). After

iodine adsorption, the two corresponding peaks shifted to 399.33 and 399.25 eV, respectively. In the case of **SUPE–py–Imine–Cage**, the C–N bond of imine peaks shifted from 399.54 eV to 400.25 eV. Meanwhile, two new peaks at 401.50 eV (for I$_2$@**SUPE–py–Imine–Cage**) and 400.69 eV (for I$_2$@**SUPE–py–Amine–Cage**) appeared attributed to the formation of charge-transfer species between the cages and iodine. Furthermore, two peaks of *O 1s* at 530.85 eV (C–O bond of the carbonyl) and 531.85 eV (trace water), respectively, were seen in the case of **SUPE–py–Imine–Cage** alone. After iodine uptake, these two peaks shifted to 531.23 eV and 532.20 eV, separately, indicating the interactions between the carbonyl units and iodine (Supplementary Fig. 17)[59,60]. Analogous results were also observed in the case of iodine adsorption using **SUPE–py–Amine–Cage** (Supplementary Fig. 17). The interactions between I$_2$ and **SUPE–py–Imine–Cage** or **SUPE–py–Amine–Cage** were further confirmed by $^1$H NMR spectroscopy performed in CDCl$_3$ (Supplementary Figs. 18–21). These findings led us to conclude that both as-prepared superphane-based cages, viz. **SUPE–py–Imine–Cage** and **SUPE–py–Amine–Cage**, are able to be used as solid sorbents for effective iodine vapor adsorption.

The excellent performance of **SUPE–py–Imine–Cage** and **SUPE–py–Amine–Cage** in iodine adsorption encouraged us to study their porosity and microstructures. To assess the porosity of these two organic cages under study, nitrogen adsorption, and desorption experiments were carried out at 77 K. As usual, the as-prepared powder was pretreated for 12 h at 100 °C under high vacuum before the N$_2$ adsorption experiments. Consequently, the Brunauer–Emmett–Teller (BET) surface areas ($S_{BET}$) were determined to be 7.5 and 13.2 m$^2$ g$^{-1}$ for **SUPE–py–Imine–Cage** and **SUPE–py–Amine–Cage**, respectively, indicating that both as-prepared cage powders are nonporous molecular materials (Fig. 3e)[61–66]. Notably, other experimental studies are needed to fully ascertain this point since the previous examples have shown that BET alone may not be enough to fully state the absence of

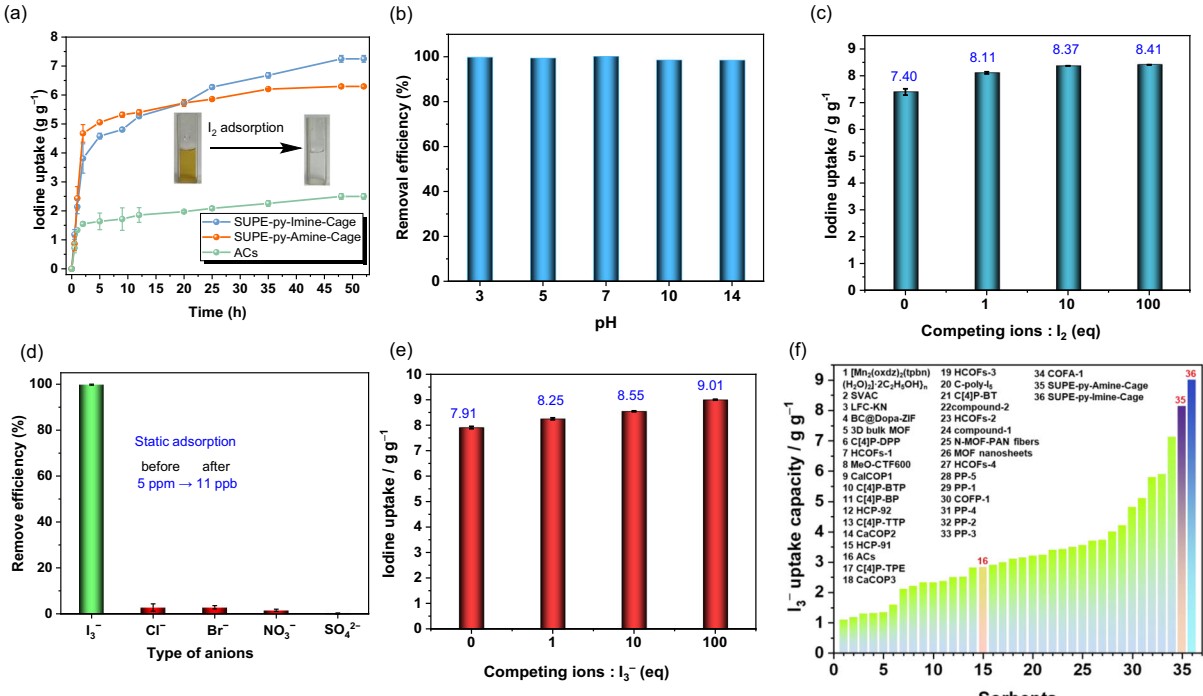

**Fig. 4 | Static iodine (I₂/I₃⁻) adsorption in aqueous media. a** Time-dependent aqueous $I_2$ (1.2 mM) adsorption studies of **SUPE–py–Imine–Cage, SUPE–py–Amine–Cage**, and ACs at ambient temperature and pressure. Insert: photographs showing the color change of aqueous $I_2$ solutions before and after adsorption with **SUPE–py–Imine–Cage. b** The pH effect on aqueous iodine (1.0 mM) adsorption with **SUPE–py–Imine–Cage. c** Iodine adsorption from aqueous iodine-containing solutions in the presence of 1–100-fold excess of co-existing competing anions (equal–molar $Cl^-$, $Br^-$, $NO_3^-$, and $SO_4^{2-}$). **d** Static selective iodine adsorption from contaminated water with trace $I_3^-$ (5.0 ppm). **e** Iodine uptake from aqueous iodine-containing solutions (600 mg KI and 300 mg $I_2$ in 100 mL of $H_2O$) in the presence of 1–100 equivalents of equal-molar $Cl^-$, $Br^-$, $NO_3^-$, and $SO_4^{2-}$ using **SUPE–py–Imine–Cage. f** The comparison of saturated iodine uptake capacities under aqueous static adsorption conditions for various reported materials (Note: listed are materials with aqueous iodine uptake capability of >1.0 g g⁻¹, also see Supplementary Table 4. All of the counter cations used were K⁺). Error bars represent SD. $n = 3$ independent experiments.

porosity in a solid. We next sought to explore whether these cage solids were crystalline or amorphous via powder X-ray diffraction (PXRD) analysis. As a result, only weak broadened signals were observed in the PXRD spectra of **SUPE–py–Imine–Cage** and **SUPE–py–Amine–Cage** (Fig. 3f), suggesting that the solids tested were amorphous. Taken together, we have evidenced, for the first time, that nonporous amorphous superphane-based organic cages are able to be used as high-performance sorbents for highly effective iodine vapor adsorption, which are essentially comparable to most known porous adsorbents for gaseous iodine uptake.

## Static iodine adsorption from contaminated water
The impressive performance of **SUPE–py–Imine–Cage** and **SUPE–py–Amine–Cage** in iodine adsorption from the gas phase allowed us to conjecture that they could be also effective for iodine capture from aqueous media. To test our proposition, we carried out adsorption experiments using iodine-aqueous solutions as the source phase (1.0 or 1.2 mM). Specifically, upon subjecting **SUPE–py–Imine–Cage** or **SUPE–py–Amine–Cage** solids to contact with aqueous $I_2$ solutions, fast iodine uptake from the aqueous source was seen, as inferred from Time-dependent UV/Vis spectroscopic studies (Fig. 4a). Meanwhile, after static iodine adsorption, the color of the aqueous solution changed from dark-brown to colorless (Fig. 4a inset). The iodine uptake equilibrium for **SUPE–py–Imine–Cage** and **SUPE–py–Amine–Cage** was nearly reached within 48 h, with the maximum equilibrium iodine uptake capability of 7.40 (corresponds to 44 $I_2$ per cage) and 6.31 (corresponds to 38 $I_2$ per cage), separately, in contrary to that of 2.50 g g⁻¹ by activated carbon under identical conditions (Fig. 4a).

Given the challenges (i.e., abundant interfering anions and extreme pHs) of aqueous iodine adsorption, especially, in the nuclear

industry, we next sought to explore whether these two superphane cages are able to selectively capture iodine from aqueous media under harsh conditions. Initially, batches (5.0 mg for each) of **SUPE–py–Imine–Cage** and **SUPE–py–Amine–Cage** were submerged into a series of readily prepared aqueous $I_2$ solutions (1.0 mM) with pH ranging from 3 to 14. The adsorption process was monitored by UV/Vis spectroscopy and the removal efficiency (equilibrium uptake percentage) was calculated based on the change in the UV/Vis absorbance intensity. As a result, the iodine removal efficiency for **SUPE–py–Imine–Cage** was measured to be >99% in the range of pH 3–7 and >98% at pH 7–14 (Fig. 4b) and that for **SUPE–py–Amine–Cage** was calculated to be ~100% over a wide pH range (3–14) (Supplementary Fig. 22). In sharp contrast, the removal efficiency of iodine for ACs was estimated to be ca. 60% under the same adsorption conditions. Of particular note was that both superphane cages, notwithstanding the occurrence of multiple imine bonds in **SUPE–py–Imine–Cage**, proved quite stable at pH 3–14, as inferred from ¹H NMR spectroscopy carried out in CDCl₃ (Supplementary Figs. 23–24).

To evaluate the adsorption selectivity of superphane cages toward iodine over other potential interfering anions, we performed competitive adsorption experiments employing a variety of simulated $I_2$-containing wastewater from nuclear industry comprising $Cl^-$, $Br^-$, $NO_3^-$, or $SO_4^{2-}$. Upon allowing either **SUPE–py–Imine–Cage** or **SUPE–py–Amine–Cage** to uptake iodine from the aqueous simulants at different pHs (3–10) consisting of $I_2$ and 1 to 10,000 equivalents of equal–molar competing anions, viz. $Cl^-$, $Br^-$, $NO_3^-$, and $SO_4^{2-}$, the iodine removal efficiency was seen essentially unchanged and retained ca. 100% for superphane cages, in comparison with the iodine removal efficiency of ca. 60% for ACs (Supplementary Figs. 25–43). More importantly, high iodine uptake

capability of up to 8.41 (for **SUPE–py–Imine–Cage**, corresponds to 50 $I_2$ per cage) and 7.06 g g$^{-1}$ (**SUPE–py–Amine–Cage**, corresponds to 42 $I_2$ per cage) was also observed even in the concurrent presence of 1 to 100 equivalents of equal–molar Cl$^-$, Br$^-$, NO$_3^-$, and SO$_4^{2-}$ in the $I_2$-containing aqueous simulants, in sharp contrast to that of 2.50 g g$^{-1}$ for ACs under identical conditions (Fig. 4c and Supplementary Fig. 44). Notably, under these experimental conditions, negligible adsorption of the interfering anions (viz. Cl$^-$, Br$^-$, NO$_3^-$, and SO$_4^{2-}$) occurred even if the molar ratio of the competing anions and $I_2$ reached up to 100 (Supplementary Fig. 45 and Supplementary Table 1), reflecting the unprecedented iodine adsorption selectivity. In aggregate, by virtue of superior iodine adsorption capability, to the best of our knowledge, both nonporous amorphous cages in question surpassed all known aqueous iodine-adsorbing material systems, including MOFs, COFs, POCs, and POPs (Supplementary Table 2).

In complementarity to neutral $I_2$, iodide ($I^-$) and tri-iodide ($I_3^-$) are two important anionic iodine species. In aqueous media such as iodine-containing wastewater, $I_2$ tends to generate tri-iodide ($I_3^-$), particularly in the presence of $I^-$, via the dynamic equilibrium $I_2 + I^- \rightleftarrows I_3^-$. Thus, adsorption of tri-iodide is also of importance to iodine remediation. To assess the effectiveness of these two superphane-based cages in $I_3^-$ (also shown as $I_2/I^-$) adsorption, iodine adsorption experiments were carried out by immersing the superphane cages in $KI_3$ (as 600 mg KI and 300 mg $I_2$ in 100 mL of $H_2O$) solution for 48 h, as monitored by UV-vis spectroscopy (the UV-vis spectra of $I_2$ and $I_3^-$ in water are totally different) (Supplementary Fig. 46). Much to our surprise, the iodine uptake capability was calculated to be 7.91 g g$^{-1}$ for **SUPE–py–Imine–Cage** and 6.31 g g$^{-1}$ for **SUPE–py–Amine–Cage**, separately. The resulting iodine-adsorbed mixtures after adsorption were confirmed by XPS, SEM–EDS, and infrared spectroscopy (Supplementary Figs. 47–54). In analogy to what has been seen in the case of the aqueous $I_2$ adsorption with superphane cages in a wide range of pH 3–14, both superphane-based molecular materials were found also capable of effectively capturing $I_3^-$ ($I_2$/KI) from aqueous media at pH 3–14, as opposed to the unsatisfactory performance of ACs (Supplementary Figs. 55, 56). More importantly, a large (1 to 100-fold) excess of competing anions (viz. Cl$^-$, Br$^-$, NO$_3^-$, and SO$_4^{2-}$) were ascertained to not essentially affect the $I_3^-$ adsorption, consistently giving the exceptionally high iodine uptake capability of up to 9.01 g g$^{-1}$ (corresponds to 53 $I_2$ or 34 $I_3^-$ per cage), as compared to the iodine uptake capability of 2.83 g g$^{-1}$ for ACs under the same conditions (Fig. 4e, Supplementary Figs. 57–77 and Supplementary Table 3). Taken in concert, recent years have witnessed tremendous progress in development of advanced adsorbents for aqueous iodine uptake, of which the highest iodine uptake capability was reported to be ca. 7.13 g g$^{-1}$ in the absence of any competing anions using a covalent organic framework aerogel system (Fig. 4f and Supplementary Table 4). To the best of our knowledge, **SUPE–py–Imine–Cage**, a new class of nonporous amorphous molecular materials, was found superior to all existing adsorbents for aqueous iodine uptake, with the highest uptake capability of up to 9.01 g g$^{-1}$ notwithstanding the concurrent occurrence of large amounts (up to 100-fold) of interfering anions (Fig. 4f).

Apart from $I_2$ and $I_3^-$, $I^-$ and $IO_3^-$ could also occur in the nuclear wastes, inter alia at low pHs. This prompted us to test if **SUPE–py–Imine–Cage** and **SUPE–py–Amine–Cage** were able to uptake and remove $I^-$ and $IO_3^-$ in aqueous solutions. We next performed similar adsorption of $I^-$ and $IO_3^-$ (K$^+$ as the counter cations) at different pHs (pH = 3–10) using **SUPE–py–Imine–Cage** and **SUPE–py–Amine–Cage**. As a result, in sharp contrast to the highly efficient and selective adsorption of $I_2$ or $I_3^-$ in aqueous media by **SUPE–py–Imine–Cage** and **SUPE–py–Amine–Cage**, no appreciable uptake of neither $I^-$ nor $IO_3^-$ was observed even if the absorption time was extended up to 3 h (Supplementary Fig. 78). As such, in consideration of the dynamic equilibrium $I_2 + I^- \rightleftarrows I_3^-$, the apparent

adsorption of $I_3^-$ ($I_2/I^-$) could result from the iodine adsorption in the form of either $I_2$ or $I_3^-$, instead of $I^-$. Given the fact that $I_2$ could be adsorbed from either water or hexane with high efficiency (Supplementary Fig. 79), we can conjecture that $I_2$ adsorption could dominate the highly efficient apparent tri-iodide uptake, in assistance of direct $I_3^-$ capture.

We next performed breakthrough experiment to assess the extreme iodine adsorption capability of these two superphane-based adsorbents from contaminated aqueous media containing trace $I_3^-$ (as low as 5.0 ppm). Satisfyingly, in the breakthrough experiment, over 99.8% of iodine ($I_3^-$) was removed after absorption by the **SUPE–py–Imine–Cage**, and the similar iodine absorption capability of 98.5% was seen for **SUPE–py–Amine–Cage**, in the absence of competing anions. More importantly, over 99.8% of iodine, as well as negligible interfering anions, were well-retained in **SUPE–py–Imine–Cage** and the residual iodine in the aqueous mixture was detected to be as low as 11 ppb (Fig. 4d) even in the presence of 1000-fold competing anions (equal-molar Cl$^-$, Br$^-$, NO$_3^-$, and SO$_4^{2-}$). In the case of **SUPE–py–Amine–Cage**, similar results were observed in the absence and presence of 1000-fold competing anions (equal-molar Cl$^-$, Br$^-$, NO$_3^-$, and SO$_4^{2-}$) (Supplementary Fig. 80).

## Dynamic flow-through adsorption from iodine aqueous solutions

The rapid, efficient, and highly selective iodine adsorption in the form of $I_2$ or $I_3^-$ from complex aqueous sources led us to consider that both superphane-based cages could be suitable adsorbents for use in a flow-through iodine capture setup, which is critical to practical applications of adsorbents for radioactive iodine remediation. To test this hypothesis, we carried out dynamic flow-through iodine adsorption experiments. Concretely, a glass pipette was charged with 10 mg of either **SUPE–py–Imine–Cage** or **SUPE–py–Amine–Cage** as the stationary phase. Then aqueous solutions of $I_2$ (1.0 mM, 10 mL) or $I_3^-$ (30 mg of KI and 15 mg of $I_2$ in 10 mL of $H_2O$) in the absence or presence of various competing anions were subjected to pass through the organic cage layer at an optimized flow-rate of 0.3 mL min$^{-1}$ for **SUPE–py–Imine–Cage**, and 0.1 mL min$^{-1}$ for **SUPE–py–Amine–Cage**, respectively (Supplementary Figs. 81–82). The flow rate was controlled by a syringe pump and the eluent was monitored directly by means of UV/Vis spectroscopy. For a more realistic simulation of the real environment (e.g. extreme pHs) of aqueous iodine dynamic flow-through adsorption, we next sought to explore whether these two superphane cages were able to selectively capture iodine from aqueous media under harsh conditions, such as $I_2$ or $I_3^-$ solutions with pH ranging from 3 to 14. As a result, the iodine removal efficiency for **SUPE–py–Imine–Cage** was measured to be ~100% over a wide pH range (Supplementary Fig. 83), and that for **SUPE–py–Amine–Cage** was calculated to be > 94% at pHs 3–7 and > 92% in the pH range of 7–14. Contrastingly, under the same conditions, the iodine uptake capability of ACs was estimated to be 40–45% at pH 3–14 at a flow rate of 0.5 mL min$^{-1}$ (Supplementary Figs. 83–84). Likewise, the $I_3^-$ removal efficiency for **SUPE–py–Imine–Cage** was measured to be ~100% in the range of pH 3–14 and that for **SUPE–py–Amine–Cage** was calculated to be >93% over a wide pH range (3–14) (Supplementary Fig. 85). In contrast, the iodine uptake capability for ACs was calculated to be between ~44% in the range of pH 3–5 and only ~34% at pH 7–14 (Supplementary Fig. 85). Satisfyingly, under the flow-through experimental conditions, the iodine ($I_2$) uptake capability was estimated to be 3.27 g g$^{-1}$ for **SUPE–py–Imine–Cage** and 2.83 g g$^{-1}$ for **SUPE–py–Amine–Cage**, respectively, as opposed to 1.59 g g$^{-1}$ for ACs, from aqueous $I_2$ solutions. To our surprise, the iodine uptake capability was measured to be up to 5.44 g g$^{-1}$ for **SUPE–py–Imine–Cage**, and 4.14 g g$^{-1}$ for **SUPE–py–Amine–Cage**, respectively (versus 1.89 g g$^{-1}$ for ACs), from $KI_3$ (as $I_2$ and KI) water solution.

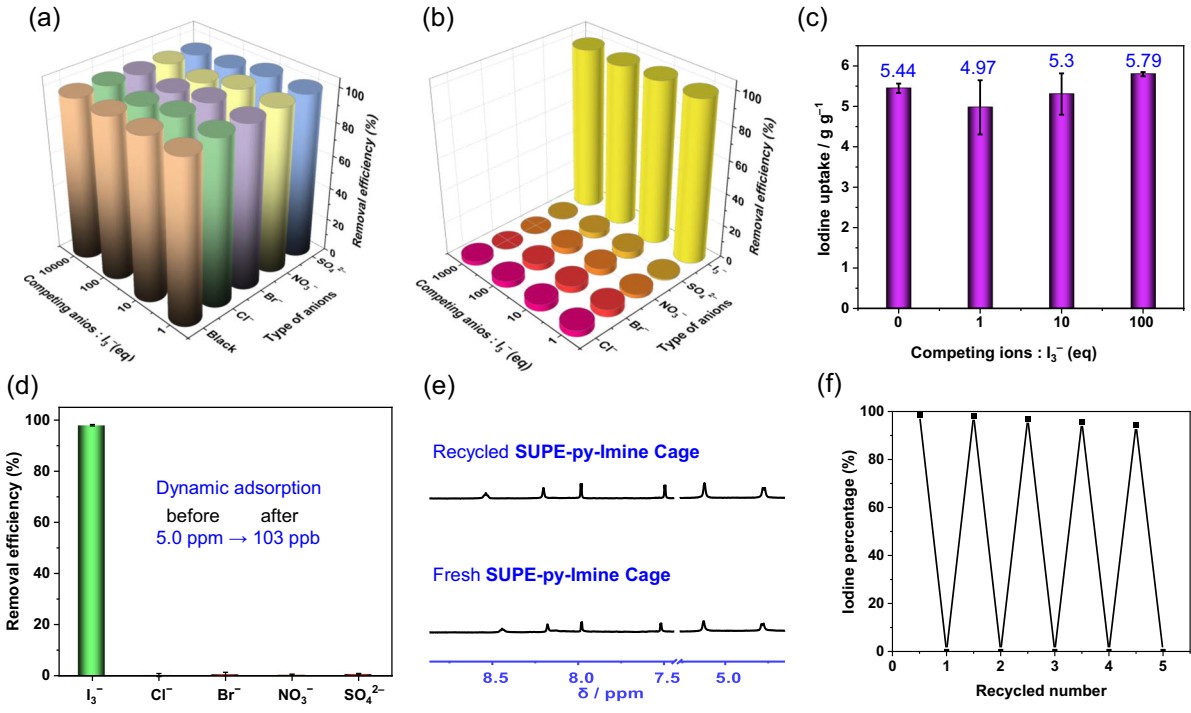

**Fig. 5 | Dynamic flow-through iodine adsorption from aqueous media with NAS materials and recyclability of the sorbents.** Dynamic removal efficiency (relative uptake) of **a** $I_3^-$ anion (60 mg KI and 30 mg $I_2$ in 10 mL of $H_2O$ for 1–100 equivalents, and 12 mg KI and 6 mg $I_2$ in 10 mL of $H_2O$ for 10,000 equivalents) in a binary aqueous mixture of competing anions and **b** anions from aqueous $I_3^-$ solutions (60 mg KI and 30 mg $I_2$ in 10 mL of $H_2O$ for 1–100 equivalents, and 12 mg KI and 6 mg $I_2$ in 10 mL of $H_2O$ for 10,000 equivalents) containing an excess (1 to 1000 equivalents) of equal-molar competing anions, viz. $Cl^-$, $Br^-$, $NO_3^-$, and $SO_4^{2-}$, with **SUPE−py−Imine−Cage**. **c** Dynamically captured amounts of iodine in the flow-through experiments using **SUPE−py−Imine−Cage** from $I_3^-$-containing aqueous mixtures (600 mg KI and 300 mg $I_2$ in 100 mL of $H_2O$) with 1 to 100 equivalents of equal-molar $Cl^-$, $Br^-$, $NO_3^-$, and $SO_4^{2-}$. **d** Dynamic iodine adsorption of $I_3^-$ from contaminated water with trace $I_3^-$ (5.0 ppm) using **SUPE−py−Imine−Cage**. **e** Partial $^1H$ NMR spectra of fresh (bottom) and recycled (top) **SUPE−py−Imine−Cage** in $CDCl_3$ after flow-through iodine adsorption. **f** The iodine removal efficiency with recycled adsorbent **SUPE−py−Imine−Cage** during 5 cycles in the dynamic adsorption experiments. (Note: all of the countercations were $K^+$). Error bars represent SD. $n = 3$ independent experiments.

The dynamic iodine removal efficiency and selectivity were further assessed with additional detailed flow-through experiments. Again, $Cl^-$, $Br^-$, $NO_3^-$, and $SO_4^{2-}$ were utilized as the competing anions to simulate real-world scenarios, such as wastewater from nuclear industry. After a binary aqueous mixture (10 mL) of $I_2$ (1.0 mM) or $I_3^-$ (30 mg of KI and 15 mg of $I_2$ in 10 mL of $H_2O$) and (1 to 10,000 equivalents of) $Cl^-$, $Br^-$, $NO_3^-$, or $SO_4^{2-}$ was passed through a pipette with 10 mg of **SUPE−py−Imine−Cage** (at a flow rate of 0.3 mL min$^{-1}$) or **SUPE−py−Amine−Cage** (at a flow rate of 0.1 mL min$^{-1}$), the iodine removal efficiency was measured to be > 95% for $I_3^-$ and > 94% for $I_2$ (versus ~40% for ACs) (Fig. 5a and Supplementary Figs. 86–87). These values were almost the same with the removal efficiency in the absence of any anions. More importantly, in the case of more complex iodine-containing aqueous mixtures simultaneously comprising 1 to 1000 equivalents of equal-molar $Cl^-$, $Br^-$, $NO_3^-$, and $SO_4^{2-}$, after passing through the solid cage layer, the removal efficiency of iodine was tested to be up to 98% for $I_3^-$ and 95% for $I_2$ with co-removal of trace competing anions, which were almost not affected by pH (from 3 to 10) (Supplementary Tables 5 and 6, Fig. 5b and Supplementary Figs. 88–90). Under the latter conditions, notwithstanding the presence of a large (up to 1000-fold) excess of interfering co-anions, the iodine uptake capability of **SUPE−py−Imine−Cage** was estimated to be 3.51 g g$^{-1}$ for $I_2$ and 5.79 g g$^{-1}$ for $I_3^-$, respectively, while that of **SUPE−py−Amine−Cage** was tested to be 2.91 g g$^{-1}$ for $I_2$ and 4.52 g g$^{-1}$ for $I_3^-$, separately, as opposed to 1.06 g g$^{-1}$ for $I_2$ and 1.11 g g$^{-1}$ for $I_3^-$ using ACs (Fig. 5c, and Supplementary Figs. 91–92).

The breakthrough curves were then obtained from the adsorption of aqueous $I_2$ or $I_3^-$ on **SUPE−py−Imine−Cage, SUPE−py−Amine−Cage**, or ACs in a fixed-bed column. Herein, we defined the value of $C/C_0$ at 0.05 as the breakthrough point, where $C_0$ is the initial concentration of sorbate (mg/L), C is the desired concentration of sorbate at time t (mg/L). At the breakthrough point, 95% removal efficiency for iodine in water was achieved. As a result, the breakthrough volume of **SUPE−py−Imine−Cage** for iodine removal was estimated to be 60 mL for $I_2$ and 3 mL for $I_3^-$, respectively. Notably, the latter smaller breakthrough volume could be attributed to the much higher initial concentration of $I_3^-$ than that of $I_2$ (3000 mg/L VS ~ 300 mg/L). Similarly, the breakthrough volume of **SUPE−py−Amine−Cage** was tested to be 40 mL for $I_2$ and 1.5 mL for $I_3^-$, separately (Supplementary Fig. 93). In contrast, the breakthrough volume for ACs was estimated to be 6 mL for $I_2$ and 1 mL for $I_3^-$, respectively, under the same conditions (Supplementary Fig. 93). In aggregate, these findings allowed us to suggest that **SUPE−py−Imine−Cage** and **SUPE−py−Amine−Cage** have much higher adsorption affinity toward $I_2$ or $I_3^-$ in aqueous solutions than commercially available ACs. We then performed breakthrough experiment to evaluate the extreme dynamic iodine adsorption capability of these two superphane-based adsorbents from contaminated aqueous media containing trace $I_3^-$ (5.0 ppm) in the presence of 1000-fold competing anions (equal-molar $Cl^-$, $Br^-$, $NO_3^-$, and $SO_4^{2-}$). Satisfyingly, after the breakthrough experiment, over 98% of iodine, as well as negligible interfering anions, were well−retained in **SUPE−py−Imine−Cage** and the residual iodine in the aqueous mixture was measured to be 103 ppb (Fig. 5d). In the case of **SUPE−py−Amine−Cage**, similar results were also observed (Supplementary Fig. 94). Again, serving as the nonporous amorphous superadsorbents, these two superphane-based cages represent the best adsorption materials for iodine uptake under the dynamic flow-through adsorption conditions.

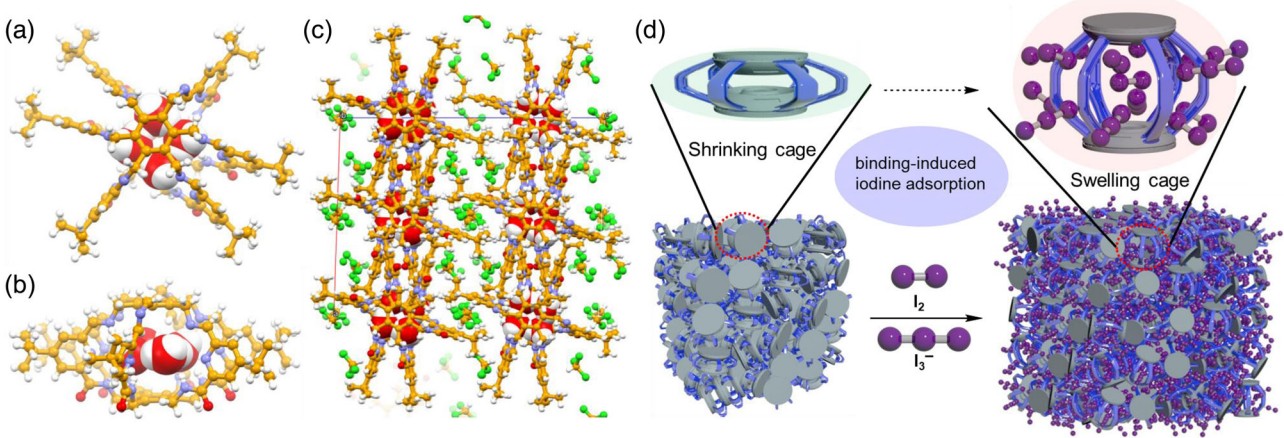

**Fig. 6 | Mechanistic study of the iodine adsorption with NAS materials.** Single-crystal structure of 4H$_2$O ⊂ **7**. **a** Top view and **b** front view. **c** The coordination network in this complex with water and chloroform molecules filling in the lattice. **d** Schematic representation of the potential working mechanism for iodine uptake.

## Recyclability, reusability, and mechanistic study

In general, recyclability and reusability of sorbents are essential for sustainable and cost-effective practical applications. The reversible iodine uptake and release are critical to the recyclability of adsorbents used. In the case of our current systems, after the iodine-adsorbed **SUPE–py–Imine–Cage** or **SUPE–py–Amine–Cage** was soaked in an isopropanol solution with sonication for 60 min, the resulting solids were filtered off, then subjected to repeat the treatment once again. As a result, the recycled solids were determined to be pure **SUPE–py–Imine–Cage** or **SUPE–py–Amine–Cage**, as confirmed by ¹H NMR spectroscopic analysis (Fig. 5e and Supplementary Fig. 95–99). This permitted us to suggest that both superphane-based cages are very stable and can be recycled by simple treatment with isopropanol under sonication, which proved more effective than conventional methods using temperature and pressure (Supplementary Fig. 100). Interestingly, iodine was found able to be released from the organic cages during the isopropanol treatment with the iodine release efficiency of 98% for **SUPE–py–Imine–Cage** and 96% for **SUPE–py–Amine–Cage**, respectively (Supplementary Figs. 101, 102). We next sought to test whether the recycled cages obtained by isopropanol-induced iodine release could be reused without performance loss. After the iodine was released from the superphane cage materials via treatment with isopropanol and sonication, the recycled cages were subjected to gaseous iodine adsorption, static iodine uptake, and dynamic flow-through iodine adsorption from contaminated water solution. The resultant performance was estimated to be almost identical to that of the fresh as-synthesized superphane cages (Fig. 5f and Supplementary Figs. 103–105). This process can be repeated 5 times and no appreciable performance loss was observed, indicating the exceptional recyclability and reusability of such type of organic cages (Fig. 5f and Supplementary Figs. 103–105).

The exceptional performance of both **SUPE–py–Imine–Cage** and **SUPE–py–Amine–Cage** on both static and dynamic iodine adsorption from contaminated water encouraged us to shed light on the working mechanism of this class of nonporous amorphous superadsorbents (NAS). Initially, the as-synthesized **SUPE–py–Imine–Cage** and **SUPE–py–Amine–Cage** were subjected to thermogravimetric analysis (TGA). In a temperature-ramped TGA measurement, the cages lost ca. 10% of their mass for **SUPE–py–Imine–Cage** and ca. 5% for **SUPE–py–Amine–Cage** between 25 and 100 °C, indicating that both cages are able to trap small guest species, e.g., H$_2$O (Supplementary Fig. 106). To get more details for supporting this notion, we carefully carried out crystallographic analysis with both super-phane cages. Luckily, suitable single crystals were obtained by vapor diffusion of tetrahydrofuran (THF) into a CHCl$_3$ solution for **SUPE–py–Imine–Cage** or via slow evaporation of a chloroform solution for **SUPE–py–Amine–Cage**. The resulting single-crystal structure of **SUPE–py–Imine–Cage** revealed that a water tetramer (4H$_2$O) was encapsulated right within the internal cavity of the lantern-like superphane cage (Fig. 6a, b). Moreover, chloroform molecules were seen to occupy the outer space of the cage as suggested by the coordination network in the complex (Fig. 6c). In the case of the single-crystal structure of **SUPE–py–Imine–Cage**, an alternate chloride hydrate tetramer was captured within the central cavity of the cage while chloroform molecules were observed to reside in the inter-molecular voids (Supplementary Figs. 107, 108). These observations lent credence to the conclusion that the superphane cages (either **SUPE–py–Imine–Cage** or **SUPE–py–Amine–Cage**) could offer both intramolecular and intermolecular voids for uptake of small guest species, e.g., H$_2$O and CHCl$_3$. Theoretically, decent to large porosity of these cage samples was expected to occur after pre-activation of the samples for 12 h at 100 °C under high vacuum before the N$_2$ adsorption experiments. Unexpectedly, negligible porosity was observed for both **SUPE–py–Imine–Cage** and **SUPE–py–Amine–Cage** according to the N$_2$ adsorption experiments (Fig. 3e). This observation could be rationalized by the fact that, upon removal of the small guests from the solids (material activation), the internal cavities might collapse and the intermolecular voids could be occupied by the tert-butyl groups of the adjacent superphane molecules (Supplementary Figs. 109, 110). Upon exposing to iodine (I$_2$ or I$_3^-$), the superphane cages are expected to bind iodine through multiple noncovalent interactions, e.g., hydrogen bonding and charge-transfer interactions between the polarized iodine and positively charged N=C bonds (Fig. 3d, Supplementary Fig. 15–21). As a result, the ingress of the iodine into the internal cavity of the cage via iodine binding could induce the swelling of the cages. Meanwhile, the swelling of the cages could further generate more intermolecular voids for trapping more iodine (Fig. 6d). This is reminiscent of a balloon-blowing event, where limited voids are found both in and between the flat balloons but large voids can be seen both in and between the blowing ones (Supplementary Fig. 111). Finally, the iodine uptake by both superphane cages can be supported by gas-phase molecular dynamics simulation, wherein the randomly distributed iodine molecules were gradually adsorbed by either **SUPE–py–Imine–Cage** or **SUPE–py–Amine–Cage** through both intramolecular and intermolecular binding (Supplementary Figs. 112–115).

In this work, we report two new superphane-based purely covalent organic cages, viz. **SUPE–py–Imine–Cage** and **SUPE–py–Amine–Cage**, with an amide unit, a pyridinyl fragment, and an imine or secondary-

amine moiety integrated on each connecting bridge of the cages. Dynamic self-assembly of a key hexakis-aldehyde precursor and hexakis (aminomethyl)benzene led to the formation of **SUPE−py−Imine−Cage**, which was efficiently reduced with NaBH$_4$ to afford its amine-version cage (**SUPE−py−Amine−Cage**). Although both organic cages were evidenced to be nonporous and amorphous, they were found able to effectively adsorb gaseous iodine with high uptake capability (up to 6.02 g g$^{-1}$). Furthermore, both **SUPE−py−Imine−Cage** and **SUPE−py−Amine−Cage** were capable of statically adsorbing I$_2$ and I$_3^-$ from aqueous media with exceptionally high uptake capability (up to 9.01 g g$^{-1}$), despite the presence of large (up to 1000-fold) excess competing anions (viz. Cl$^-$, Br$^-$, NO$_3^-$, and SO$_4^{2-}$). More importantly, in the dynamic flow-through experiments, these two superphane cages exhibited record-breaking I$_2$ and I$_3^-$ uptake capability of up to 5.79 g g$^{-1}$ from aqueous mixtures consisting of abundant competing anions. To the best of our knowledge, **SUPE−py−Imine−Cage** represents the best adsorbent materials for aqueous iodine adsorption not only under static adsorption conditions but also in the dynamic flow-through setup. Of particular note is that both cages can be utilized for efficiently and rapidly removing trace iodine in aqueous media from ppm level to ppb level (as low as 11 ppb), with excellent recyclability and reusability. We thus suggest that discrete macrocycles and cages alike, regardless of their porous or crystalline states, could serve as novel high-performance nonporous amorphous molecular materials for uses in, for instance, gas storage, mining, and wastewater remediation via effective and selective guest adsorption and separation. More work on superphane-based NAS materials is currently in progress in our laboratory.

## Methods

### General information

All solvents and chemicals were purchased in analytical purity from J&K, TCI, Energy-Chemical, or Acros and used without further purification. TLC analyses were carried out using Sorbent Technologies silica gel (200 mesh) sheets. Flash column chromatography was performed on silica gel (300–400 mesh). $^{1}$H and $^{13}$C NMR spectra were recorded on Bruker AVANCE 400 spectrometers and the spectroscopic solvents were purchased from Cambridge Isotope Laboratories or Sigma-Aldrich. Either residual solvent peak or tetramethylsilane (TMS) was used as an internal reference. The chemical shifts are expressed in δ (ppm). High-resolution mass spectra (HRMS) were recorded on a Bruker Apex-Q IV FTMS mass spectrometer using ESI (electrospray ionization) employing a CH$_3$OH as the solvent. X-ray crystallographic analyses were carried out on a Bruker D8 Venture diffractometer using a μ-focused Cu−Kα radiation source (λ = 1.5418 Å) or Agilent SuperNova system equipped with a mirror monochromator and a Cu−Kα INCOATEC IμS microfocus source (λ = 1.5418 Å). All DFT calculations were carried out with the Gaussian 16 suite[67] of programs using the X3LYP density functional[68]. Structural optimization was performed using a 6−31 G* basis set[69,70]. All molecular dynamics simulations were carried out using Gromacs 2018.8 program with general Amber force fields (GAFF)[71]. All the molecules were put into a cubic box with a side length of 6 nm using packmol software[72] (50 I$_2$ molecules for 1 imine-based cage and 42 I$_2$ molecules for 1 secondary-amine-based cage molecule). The cutoff for neighbor list of Verlet method and that for short-range interactions is 0.9 nm in all calculation with periodic boundary conditions in all three directions. After the energy minimization, the systems were run in NVT ensemble for 1 ns at 298.15 K. The time step of each simulation was 1 fs.

Deconvolution of XPS spectra was usually applied to process low energy-resolution data for possible comparison with data collected at high resolution. The raw XPS spectral data were firstly baseline-corrected and were analyzed using a peak-fitting method. This involved assigning initial peak positions based on expected binding energy values for the elements of interest, along with expected peak

widths based on typical chemical shifts compared with the reported data in literature. The fitting process was iterative, which involved adjusting the peak parameters (such as position, width, and shape) until the best fit to the original data was achieved, as determined by a least-squares fitting algorithm.

### Synthesis of SUPE−py−Imine−Cage (7)

1,2,3,4,5,6-benzenehexamethanamine **4** (0.31 g, 1.20 mmol) and **6** (1.00 g, 0.82 mmol) were dissolved in a mixture of CH$_2$Cl$_2$ and CH$_3$OH (1:1, v/v, 800 mL) and the mixture was stirred overnight at 60 °C. After the solution was cooled to room temperature, the solvent was removed under reduced pressure. The resulting residue was soaked with a large amount of H$_2$O under sonication for 10 min. The solids were filtered and washed with water, dried under vacuum to give 766 mg of **SUPE−py−Imine−Cage** (**7**) as a yellowish solid in 61% yield. $^{1}$H NMR (400 MHz, CDCl$_3$) δ 8.42 (s, 6H), 8.15 (s, 6H), 7.96 (d, J = 1.9 Hz, 6H), 7.49 (d, J = 1.9 Hz, 6H), 5.10 (s, 12H), 4.76 (d, J = 4.8 Hz, 12H), 1.11 (s, 54H). $^{13}$C NMR (100 MHz, CDCl$_3$) δ 163.2, 163.2, 162.1, 151.8, 149.8, 139.2, 137.9, 121.4, 121.0, 57.9, 38.6, 35.1, 30.5. HRMS (ESI) *m/z*: [M + H]$^+$calcd for C$_{90}$H$_{103}$N$_{18}$O$_6^+$ 1531.8308, found 1531.8302.

### Synthesis of SUPE−py−Amine−Cage (8)

500 mg of **SUPE−py−Imine−Cage** (**7**) was dissolved in a mixture of CH$_2$Cl$_2$ and CH$_3$OH (1:1, v/v, 300 mL), and NaBH$_4$ (1.73 g, 45.6 mmol) was added in portions and then stirred at room temperature overnight. The precipitates were filtered off and the filtrate was concentrated under reduced pressure. The resulting residue was soaked with a large amount of H$_2$O under sonication for 30 min. The solid residue was filtered and washed with water, dried under vacuum to give 483 mg of secondary-amine-based superphane **SUPE−py−Amine−Cage** (**8**) as yellowish solid in 87% yield. $^{1}$H NMR (400 MHz, CDCl$_3$) δ 8.23 (s, 6H), 7.93 (s, 6H), 7.17 (s, 6H), 4.75 (s, 12H), 4.13 (s, 12H), 3.41 (s, 12H), 1.33 (s, 54H). $^{13}$C NMR (101 MHz, CDCl$_3$) δ 163.2, 163.2, 162.1, 151.8, 149.8, 139.2, 137.9, 121.4, 121.0, 57.9, 38.6, 35.1, 30.5. HRMS (ESI) *m/z*: [M + H]$^+$calcd for C$_{90}$H$_{115}$N$_{18}$O$_6^+$ 1543.9241, found 1543.9275.

### Iodine vapor uptake capacity

Experiments on the adsorption of iodine vapor were determined by gravimetric measurements. The adsorbents were used as prepared or pre-activated at 348.15 K for 24 h. Then, 10 mg of adsorbent was weighed in small weighing vials, which were located in a sealed container with iodine pellets kept at the bottom. The container was placed under 348.15 K for adsorption and the vials containing residual adsorbent were weighed over different time periods.

The amount of adsorbed iodine was determined using the following equation:

$$G_t = \frac{g_t - g_0}{g_0}$$

where $G_t$ (g g$^{-1}$) is the amount of iodine-adsorbed per gram of adsorbent at time *t* (min). $g_0$ (mg) and $g_t$ (mg) are the initial and residual weight of the vials containing the adsorbent, respectively.

### Iodine adsorption efficiency from aqueous media

The adsorbent (5 mg) was immersed in an aqueous iodine solution (I$_2$: 1.0 mM, 5 mL, or I$_3^-$ 15 mg KI and 7.5 mg I$_2$ in 5 mL of H$_2$O for 1–100 equivalents, and I$_2$: 0.5 mM, 5 mL, or I$_3^-$ 3 mg KI and 1.5 mg I$_2$ in 5 mL of H$_2$O for 1000 or 10,000 equivalents in the absence or presence of competing anions (viz. Cl$^-$, Br$^-$, NO$_3^-$, or SO$_4^{2-}$). The aqueous solution was monitored by the UV-Vis spectroscopy and ion chromatography. The iodine removal efficiency (%) of the corresponding adsorbent was

determined using the following equation:

$$\text{Iodine removal efficiency} = \frac{C_0 - C_t}{C_0} \times 100\%$$

where $C_O$ (mM) and $C_t$ (mM) are the concentrations of aqueous iodine before and after adsorption, respectively.

## Others adsorption efficiency in solutions

The adsorbent (5 mg) was immersed in an aqueous I⁻ or IO$_3$⁻ solution (5 mM, 5 mL). The aqueous solution was monitored by the ion chromatography.

The removal efficiency (%) of the corresponding adsorbent was determined using the following equation:

$$\text{Removal efficiency} = \frac{C_0 - C_t}{C_0} \times 100\%$$

where $C_O$ (mM) and $C_t$ (mM) are the concentrations of aqueous iodine before and after adsorption, respectively.

The adsorbent (5 mg) was immersed in $I_2$ solution in n-hexane (5 mM, 5 mL). The solution was monitored by the UV-Vis spectroscopy.

The iodine removal efficiency (%) of the corresponding adsorbent was determined using the following:

$$\text{Iodine removal efficiency} = \frac{C_0 - C_t}{C_0} \times 100\%$$

where $C_O$ (mM) and $C_t$ (mM) are the concentrations of aqueous iodine before and after adsorption, respectively.

## Iodine adsorption capability from aqueous media

The adsorbent was immersed in an aqueous iodine solution ($I_2$ (adsorbent: 3 mg): 1.2 mM, 100 mL, or $I_3$⁻ (adsorbent: 5 mg): 600 mg KI and 300 mg $I_2$ in 100 mL of $H_2O$) in the absence or presence of competing anions (viz. Cl⁻, Br⁻, NO$_3$⁻, or SO$_4$²⁻) stirred at room temperature for 48 h. The mixture was filtrated, washed with water until the filtrate become clear. To verify the $I_2$ and $I_3$⁻ adsorption efficiency obtained from the titration analysis. The filtrate was added a 2% starch indicator aqueous solution (2 mL), and then the mixture solution was titrated by sodium bisulfite aqueous solution (0.05 mol L⁻¹) dropwise until the solution color turned from blue to transparent. The adsorption efficiency of $I_2$ or $I_3$⁻ was calculated from the quantity of sodium bisulfite aqueous solution required[19,73].

## Trace iodine adsorption efficiency from aqueous solutions

In order to measure the trace $I_3$⁻ adsorption efficiency of an aqueous solution in the absence or presence of competing anions (viz. Cl⁻, Br⁻, NO$_3$⁻, and SO$_4$²⁻), the adsorbent (10 mg) was soaked in an $I_3$⁻ aqueous solution (5 ppm, 10 mL) stirred at room temperature for 24 h. To verify the $I_3$⁻ adsorption efficiency obtained from the gravimetric analysis, the iodine loss in the filtrate was also measured. To start, a 2% starch indicator aqueous solution (2 mL) was added to the combined filtrate, and then the solution was titrated by adding 0.1 mmol L⁻¹ sodium bisulfite aqueous solution dropwise until the solution color turned from blue to transparent. The adsorption efficiency of $I_3$⁻ was calculated from the quantity of sodium bisulfite aqueous solution required.

## Adsorption kinetics

The adsorption kinetics were quantified using Ho and McKay's pseudo-second-order model from which the apparent rate constant $k_{obs}$ can be obtained:

$$t/q_t = t/q_e + 1/(k_{obs} \times q_e^2)$$

where $q_t$ and $q_e$ are the adsorbate uptakes (g adsorbate per g adsorbent) at time $t$ (min) and at equilibrium, respectively, and $k_{obs}$ is an apparent second-order rate constant (g g⁻¹ min⁻¹). The rate constant $k_{obs}$ can be calculated from the intercept and slope of the plot of $t/q_t$ against t.

## Dynamic flow-through adsorption efficiency

A column with a cross-sectional area of 4 mm² was filled with adsorbent (10 mg). Then an aqueous iodine solution ($I_2$: 1.0 mM, 10 mL, or $I_3$⁻: 60 mg KI and 30 mg $I_2$ in 10 mL of $H_2O$ for 1–100 equivalents, and $I_2$: 0.5 mM, 10 mL, or $I_3$⁻ 12 mg KI and 6 mg $I_2$ in 10 mL of $H_2O$ for 1000 or 10,000 equivalents) in the absence or presence of competing anions (viz. Cl⁻, Br⁻, NO$_3$⁻, or SO$_4$²⁻) was passed through this column at a flow rate of 0.3 mL min⁻¹ for **SUPE–py–Imine–Cage**, 0.1 mL min⁻¹ for **SUPE–py–Amine–Cage** and 0.5 mL min⁻¹ for ACs as controlled by a syringe pump. The eluent was analyzed directly by means of UV/Vis spectroscopy, and the competing anions in the solution were tested by ion chromatography. The efficiency of iodine (%) by the adsorbent was determined using the following equation:

$$\text{Dynamic flow-through adsorption efficiency} = \frac{C_0 - C_t}{C_0} \times 100\%$$

where $C_O$ (mM) and $C_t$ (mM) are the concentrations of iodine in the aqueous solution before and after adsorption, respectively.

## Dynamic flow-through adsorption capacity

A column with a cross-sectional area of 4 mm² was filled with adsorbent (10 mg). An aqueous iodine solution ($I_2$: 1.2 mM, 200 mL, or $I_3$⁻: 600 mg KI and 300 mg $I_2$ in 100 mL of $H_2O$) in the absence or presence of competing anions (viz. Cl⁻, Br⁻, NO$_3$⁻, and SO$_4$²⁻) was then passed through the column at a flow rate of 0.3 mL min⁻¹ for **SUPE–py–Imine–Cage**, 0.1 mL min⁻¹ for **SUPE–py–Amine–Cage** and 0.5 mL min⁻¹ for ACs as controlled by a syringe pump. To test the $I_2$ and $I_3$⁻ adsorption efficiency obtained from the titration analysis, the residual liquid was added a 2% starch indicator aqueous solution (2 mL), and then the mixture solution was titrated by sodium bisulfite aqueous solution (0.05 mol L⁻¹) dropwise until the solution color turned from blue to transparent. The adsorption efficiency of $I_2$ or $I_3$⁻ was calculated from the quantity of sodium bisulfite aqueous solution required.

## The breakthrough curve of dynamic flow-through adsorption

A column with a cross-sectional area of 4 mm² was filled with adsorbent (10 mg). An aqueous iodine solution ($I_2$: 1.2 mM, 200 mL, or $I_3$⁻: 600 mg KI and 300 mg $I_2$ in 100 mL of $H_2O$) in the absence or presence of competing anions (viz. Cl⁻, Br⁻, NO$_3$⁻, and SO$_4$²⁻) was then passed through the column at a flow rate of 0.3 mL min⁻¹ for **SUPE–py–Imine–Cage**, 0.1 mL min⁻¹ for **SUPE–py–Amine–Cage** and 0.5 mL min⁻¹ for ACs as controlled by a syringe pump. The adsorption efficiency of the $I_2$ and $I_3$⁻ was obtained by the UV/Vis spectroscopy, and breakthrough curve was obtained by plotting C/C$_0$ (as $y$ axis) against elution volume (as $x$ axis). The breakthrough point was defined as the C/C$_0$ value of 0.005, where $C_O$ (mM) and C (mM) are the iodine concentrations of solutions at the Inlet and the outlet, respectively.

## Regeneration and recycling experiment

$I_2$ or $I_2$/KI-loaded samples were subjected to sonication in iPrOH (10 mL) for 60 min. The color of solution was rapidly turned brown over time. This process was monitored using time-dependent UV-Vis spectroscopy and repeated 5 times. The purity of the recycled cages was further confirmed by ¹H NMR spectroscopy performed in CDCl$_3$ or CDCl$_3$/CD$_3$OD (1/1, v/v).

## Data availability

All the data supporting plots within this study are included in this article and its Supplementary files. Source data are provided by the corresponding authors upon request. X-ray structural data for super-phane cages **SUPE–py–Imine–Cage (7)** and **SUPE–py–Amine–Cage (8)** have been deposited at the Cambridge Crystallographic Data Center, under deposition numbers CCDC 2245537 and 2244538. Copies of the data can be obtained free of charge via https://www.ccdc.cam.ac.uk/structures/. Energies and geometrical coordinates of the optimized models in the gas phase for **SUPE–py–Imine–Cage** are included in the supplementary information.

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

## Acknowledgements

This research was funded by the National Natural Science Foundation of China (22071050 to Q.H.), the Science and Technology Plan Project of Hunan Province, China (Grant no. 2019RS1018 to Q.H.), and Fundamental Research Funds for the Central Universities (Startup Funds to Q.H.). We thank Dr. Zhenyi Zhang from Bruker (Beijing) Scientific Technology Co., Ltd for helpful discussions on X-ray crystallography.

## Author contributions

Conceptualization and supervision: Q.H.; synthesis: W.Z., M.Z., and Y.X.; characterization, NMR, and iodine adsorption experiments: W.Z.; single-crystal growing, data collection and analysis: W.Z. and Q.H.; theoretical calculations: A.L.; thermogravimetric analysis: Y.Z.; writing–original draft: W.Z. and QH; reviewing and editing: Q.H. All authors proofread, commented on, and approved the final version of this manuscript.

## Competing interests
