## [Peer Review File · Nature Communications]

Nonporous Amorphous Superadsorbents for Highly Effective and Selective Adsorption of Iodine in WaterREVIEWER COMMENTS

Reviewer #1 (Remarks to the Author):

In this manuscript entitled " Nonporous Amorphous Superadsorbents for Highly Effective and Selective Adsorption of Iodine in Water", the authors synthesized and reported an amorphous superphane (SUPE-py-Imine-Cage) and its reduced product, another amorphous superphane (SUPE-py-Amine-Cage). Both of cages exhibited high iodine uptake in vapor media and especially, showed excellent affinity and exceptional uptake towards iodine in solution media. The possibility of application under practical scenario was validated by dynamic flow-through experiments of two superphane-based sorbents and the dynamic adsorption data is attractive. The mechanism is supported by XPS and NMR results. Nevertheless, several issues are listed below that I think the authors should address before this manuscript could achieve the standard of this journal.

1. Nonporous amorphous superphane-based cages in this work are interesting, as they exhibited exceptional capacity for iodine compared to conventional nonporous amorphous sorbents and the authors here call it "game changers" with this logic, a conceptual point combined . Unfortunately, there are two comments they ignored: First, as they deemed, the amorphous nature of this two cages may provide more stretchable space, leading to high capacity for iodine molecules but to the best of my knowledge, this type of superphane-based cages are orderly crystals in similar reported studies (e.g., Cell Reports Physical Science, 2023, 4, 101295; Chem. Commun., 2021, 57, 4496-4499), through which the host-guest interaction can be tailored specifically. In contrast the amorphous cages are unable to achieve it in this work. Second, unlike other guest anions, I₃⁻ is special due to its complex properties as it not only can be recognized and trapped by cavity based on I⁻, but can be adsorbed on the surface of sorbent based on I₂ and formation of charge-transfer complexes. That's to say the authors are supposed to clarify and give more discussion about the assistance toward iodine capture from the amorphous structure, otherwise the philosophy and presume of the sorbent are unable to be hold, just originating from their synthesis and single crystal culturing remained to be improved (in fact they obtained single crystals of one of the cages).

2. In Supplementary Fig. 7, the characteristic peaks attributed to imine-N and pyridine-N

ought to be divided before and after iodine adsorption.

3. In Supplementary Fig. 8, the annotations corresponding to non-I₂-loaded and I₂-loaded should be provided in figures.

4. In page 17, it is deemed that the I₃⁻ are combined by hydrogen bonding and charge-transfer interactions between I and N. However, especially for the hydrogen bonding, it can't be supported by current evidence in ¹H NMR spectra with low resolution. Crystal structure of I₃⁻@SUPE-py-Imine-Cage or I₃⁻@SUPE-py-Amine-Cage or use of MD simulations and ab initio molecular dynamics could support their mechanism.

5. Adsorption isotherm model for static I₃⁻ adsorption in aqueous media are supposed to be provided. Besides, the conditions of batch adsorption experiments (e.g., initial concentration of I₃⁻ and competing anions) should be addressed in caption for Fig 3 and Fig 4.

6. For dynamic adsorption, the breakthrough curve should be plotted and corresponding analysis should be provided.

7. To figure out the contribution for ionic I⁻ and molecule I₂, the supplementary adsorption experiments in solution for single I⁻ (in water) and single I₂ (in hexane) are supposed to be conducted.

Reviewer #2 (Remarks to the Author):

The manuscript entitled "Nonporous amorphous superadsorbents for highly effective and selective adsorption of iodine in water" submitted to be considered for publication in *Nat. Commun.* describes the synthesis and characterization of a hexa-imine superphane dynamic covalent cage and its covalent amine counterpart. They evaluate their use in I₂ and I₃⁻ adsorption from I₂ vapor and from aqueous sources either in batch or in flow. They claim that the prepared superphane cages have superior adsorption properties than any of the previously porous adsorbents reported to date.

The work is very similar in terms of experimental procedures and discussion to the one reported last year in *Angew. Chem. Int. Ed.* (ref 18 in the original manuscript) from Ji, Sessler, Wang and collaborators. The main differences with the previously reported work are basically the receptor used as adsorbent (superphane vs calix[4]pyrrole-based crosslinked polymer) and the superior uptake capacity described in the present work (iodine

vapor adsorption capacity 9 g g⁻¹ vs 3 g g⁻¹, respectively, and iodine capture from water 5 g g⁻¹ vs 3 g g⁻¹, respectively). The differences in uptake capacity are quite significant.

However, I would suggest additional experiments to confirm these results.

In the present work the authors only report the uptake capacities, kinetics, and recovery with the imine and amine cages prepared. They compare their results with the reported ones but they do not report any control experiment performed exactly in the same conditions they use. I strongly suggest to perform the same experiments using a typical activated carbon and compare their results under the same experimental conditions and applying the same data analysis. Having their own control experiment will discard experimental differences on the obtained data compared to described data and will provide more reliable conclusions.

The work is written in a clear manner and most of the experimental procedures are well-described. However, I suggest to clarify how did they control the equivalents of I₂ added in the NMR titration data included in the SI (S10 and S12).

Additionally, I recommend the authors to address the following issues in the revised version of the manuscript:

1) The authors use ¹H NMR spectroscopy to support the binding of iodine with imine and amine cages. Figures S9-S12 in the supporting information show different exchange kinetics in the ¹H chemical shift time scale for imine and amine cages (fast for imine and fast for amine). The authors do not mention anything about this in the main text. I suggest to add a comment on that to explain the differences observed. Is the binding mechanism somehow related to the adsorption mechanism? Please add a comment on that.

2) The authors only report the adsorption kinetics for the uptake in vapor phase. What about the studies in aqueous solution? I suggest to add also these values and compared them with a control adsorbent (activated carbon).

3) The authors claim that they can recycle the adsorption material by sonication in isopropanol. I wonder if they could simply use temperature and vacuum to recycle the material as done for previously reported materials. Please add a comment on that.

4) The authors show the iodine adsorption at different pHs 3-10. They refer always to I₃⁻ and I₂ extraction. At different pHs in aqueous media other forms of dissolved I are possible: I⁻, IO₃⁻, HOI, etc. I suggest to add a comment on how this can affect the selectivity and the adsorption capacity of the cages. Please add a comment on which as the basis of the

observed selectivity for I₂ and I₃⁻ over the interfering anions tested.

Although the reported results are of high quality, the authors need to address the issues raised above prior accepting it for publication in any journal. After addressing the issues and confirming the outstanding uptake performance I will be happy to recommend it for publication in Nat. Commun.

Reviewer #3 (Remarks to the Author):

The field of non-porous amorphous materials (yet capable of adsorption) is very new and recent with few examples. Using a simple covalent cage is attractive and opens a new paradigm because the simplicity of the approach suggests that many previously synthesized cages (overlooked) and more to come are capable of similar things and perhaps (actually probably) not only restrained to I₂ adsorption. The simplicity of avoiding ordering the matter in 3D to get a crystalline material is of huge interest notably for industry. Rest the evaluation of energetic and product cost of production and re-activation which should be mentioned as perspectives but falls beyond the scope of this fundamental research in my opinion. The exceptional performances in terms of iodine capture in harsh (pH) and competitive (anions) conditions plus the highest capacities so far from aqueous solutions are sufficient arguments, to me, to justify publication for a leadership journal of broad audience. However, I have several concerns which I believe should be addressed before publication.

1- Introduction: Please explain better in which context these 2 radionuclides ¹²⁹I and ¹³¹I are produced. Is it during normal civil nuclear energy exploitation or after a nuclear accident? (both?) Otherwise we do not understand why this is urgent to develop such materials to catch dangerous isotopes of iodine. To me it is not a present need but it could become very urgently needed in case of problems. So I appreciate the initiative to anticipate and possibly make available solutions if dramatic issues raise in the future.

2- Explain that 1st gaseous I₂ needs to be trapped by specific sorbents and then 2nd dissolved I₂ in water which is the focus of this paper.

3- When citing NACs, actually non-porous adaptive crystals are much older: Science Atwood,

2002 (Storage of Methane and Freon by Interstitial van der Waals Confinement). Please state this clearly. Other papers appeared after that and before NACs like Gas-induced transformation and expansion of a non-porous organic solid, *Nat Mater* 2008 or Diffusion of vaporous guests into a seemingly non-porous organic crystal, *ChemComm* 2014 ...

4- Page 2, “unwanted interference of water molecules”, this critical point is justly raised by the authors. I would just add this paper in the proposed reference list as a noticeable example/exception: “*Angew. Chemt. Int Ed. Energy-efficient iodine uptake by a molecular hostguest crystal, 2022*”.

5- Why perform I₂ uptake at 348K? In water or atmosphere, the temperature is likely not 348K. Please justify both in the letter and in the manuscript for readers and clarity.

6- Figure 2d and corresponding text (and later Figure 5d): there is a problem. Data suggest formation of anionic iodine species I₃⁻ and I₅⁻. However, this leaves the charge balance unequilibrated and positive species must have formed but there is no data or assumption of what happened there. Please explain or suggest a mechanistic explanation of what happened (transformation of I₂ into I₃⁻ or I₅⁻ and which cation formed to balance the charge to 0).

7- What is the reason for the specificity for I₃⁻ anions versus other anions? Please suggest a rationale.

8- It lacks a ratio between the number of cage molecules and the number of trapped I₂ and I₃⁻. For example, giving the correct numbers in the form X molecules of I₂ per cage molecule would help the reader to picture how many molecules of interest have been trapped by 1 molecule of cage. This would also help decipher the location of trapped molecules for example if relatively high, this would be an additional argument toward intermolecular association if intramolecular is privileged. This should be briefly discussed.

9- About the very few previous examples of non-porous amorphous materials yet capable of adsorption, I advise the authors to add these references which are clearly in the scope of this paper: Adsorptive Separation of Benzene, Cyclohexene, and Cyclohexane by Amorphous Nonporous Amide Naphthotube Solids, *Angew Chem Int Ed* 2020, 59, 19945-19950; *J. Phys. Chem. C* 2011, 115, 47, 23344; *Bulletin of Chemical Reaction Engineering & Catalysis*, 12 (2), 263-272 (doi:10.9767/bcrec.12.2.766.263-272; the following paper is advised to be cited for readers: *CrystEngComm*, 2012,14, 1909-1919. I also invite the researchers to very carefully read this paper (DOI: 10.1002/adfm.200800624) which should merit high attention from the

community stating that measuring a N₂ BET alone may not be sufficient to state about the absence of porosity in a material. I invite the authors to keep their statements about nonporosity but add a sentence saying something that nevertheless, previous examples have shown that BET alone may not be enough to fully state the absence of porosity in a solid and that other experiments studies are needed to fully ascertain this point. This in my opinion does not impede the relevance of the work nor decreases the quality of the material which was proven on multiple occasions to be fully relevant for iodine iodide capture in (much appreciably) efforts toward more real conditions of use).

10- Why tBu groups are present in the superphane cages? Please explain briefly why it has been introduced in the syntheses.

11- Page S5, crude formula of compound 6 is incorrect, I found C₇₈H₉₀O₁₂N₁₂ and so the MS. Please correct.

12- Supplementary Figures containing XPS spectra, please explain how the signal deconvolution was done. It can be by a simple sentence or paragraph in the general information, early in ESI.

13- Tests with anions, no salt in indicated so we do not know which cation was used, please indicate. Also explain (in the paper) the rationale for the use of these anions.

Supp Fig 16, why indicating competing anions if I₂ is considered? Please explain or correct.

Supp Fig 16 to 18 and everywhere necessary, please indicate concentration of cage and if relevant concentration of anions. Supp Fig S23 to S28, which pH was used? Following Figures, pH missing was well as cage concentration (weight conc as I guess the cages are not soluble in water or indicate which mass of cage was used for given volumes). Supp tables, how the uptake amounts of anions (particularly low) were determined?

14- Stability experiments of cages exposed to different pH are very nice. Comparisons with known compounds, very nice and insightful.

15- Something remains unclear. Why after having trapped X equiv. of I₂ or I₃⁻, these species continue to be attracted to reach such high levels of adsorption? We could rationally expect that, after having trapped a few equiv. of iodine species, the process would stop? Please try to propose a hypothesis to account for this result (internal and external adsorption, I₂-I₂ interactions or I₃⁻/I₃⁻ interactions, hydrophobic effect, etc ... which would make sense to the authors.

16- Something not so surprising is the adsorption of I₃⁻ from water as, iodine in the form of

I3- is adsorbed when cages are exposed to I2. Nevertheless, why did the authors studied I2 and I3- but not I-? They mention formation of I3- from I2 and I- so, to form I3-, I- must be present. Please provide results with I-, even if not good, results are so excellent with the two other species (I2 and I3-) that it deserves publication.

17- Instead of cavity self-filling I would talk about void or cavity collapse upon material activation, if so.

18- CCDC numbers are not given for the crystal structures.

Response to the comments of Reviewer #1:

In this manuscript entitled " Nonporous Amorphous Superadsorbents for Highly Effective and Selective Adsorption of Iodine in Water", the authors synthesized and reported an amorphous superphane (SUPE-py-Imine-Cage) and its reduced product, another amorphous superphane (SUPE-py-Amine-Cage). Both of cages exhibited high iodine uptake in vapor media and especially, showed excellent affinity and exceptional uptake towards iodine in solution media. The possibility of application under practical scenario was validated by dynamic flow-through experiments of two superphane-based sorbents and the dynamic adsorption data is attractive. The mechanism is supported by XPS and NMR results. Nevertheless, several issues are listed below that I think the authors should address before this manuscript could achieve the standard of this journal.

Reply: We do appreciate this reviewer's great and professional comments. Below are the detailed responses to this reviewer's comments and concerns.

1. Nonporous amorphous superphane-based cages in this work are interesting, as they exhibited exceptional capacity for iodine compared to conventional nonporous amorphous sorbents and the authors here call it "game changers" with this logic, a conceptual point combined. Unfortunately, there are two comments they ignored: First, as they deemed, the amorphous nature of this two cages may provide more stretchable space, leading to high capacity for iodine molecules but to the best of my knowledge, this type of superphane-based cages are orderly crystals in similar reported studies (e.g., Cell Reports Physical Science, 2023, 4, 101295; Chem. Commun., 2021, 57, 4496-4499), through which the host-guest interaction can be tall specifically. In contrast the amorphous cages are unable to achieve it in this work.

Reply: We are grateful for this reviewer's positive comments. As this reviewer may appreciate, we indeed reported single-crystal structures of other types of superphane-based molecular constructs, resulted from single crystals obtained under crystallization conditions, in our previous papers (e.g., Cell Reports Physical Science, 2023, 4, 101295; Chem. Commun., 2021, 57, 4496-4499). We then reperformed powder X-ray diffraction (PXRD) analysis with those as-synthesized solids. As a result, analogous to what was seen in this work, the as-synthesized superphane-based cage and its anionic carceplexes reported in Cell Reports Physical Science, 2023, 4, 101295 by our group are found amorphous solids as well (Supplementary Fig. R1a-c). But the as-prepared superphane cage reported in Chem. Commun., 2021, 57, 4496-4499 was somewhat seen crystalline (Supplementary Fig. R1d). As such, the as-synthesized superphane-based cages are not always orderly crystals, which depends on the crystallinity of the superphane cages per se, along with their host-guest interactions and perhaps more. We thus don't think there is a close relationship between the crystalline or amorphous nature of the cages and the strong host-guest interactions.

[Figure redacted]

Supplementary Fig. R1 | Powder X-ray diffraction patterns of as-synthesized (a) superphane **R1**; (b) ($\text{H}_2\text{PO}_4^- \text{ @R1}$); (c) ($\text{AsO}_4^{3-} \text{ @R1}$) reported in *Cell Reports Physical Science*, **2023**, 4, 101295 and (d) imine-containing superphane cage reported in *Chem. Commun.*, **2021**, 57, 4496-4499.

Second, unlike other guest anions, I_3^- is special due to its complex properties as it not only can be recognized and trapped by cavity based on I^- , but can be adsorbed on the surface of sorbent based on I_2 and formation of charge-transfer complexes. That's to say the authors are supposed to clarify and give more discussion about the assistance toward iodine capture from the amorphous structure, otherwise the philosophy and presume of the sorbent are unable to be hold, just originating from their synthesis and single crystal culturing remained to be improved (in fact they obtained single crystals of one of the cages).

Reply: We do agree with this reviewer that I_3^- is special due to its complex properties, inter alia its dynamic equilibrium $\text{I}_2 + \text{I}^- \rightleftharpoons \text{I}_3^-$. According to the iodine adsorption experiments, I_2 in aqueous solution can be efficiently adsorbed by both superphane cages. In contrast, they failed to uptake anionic I^- in water solution even if the adsorption time was extended to 3 h (Supplementary Fig. 69). As such, the actual uptake of the complex I_3^- species could processed in the form of I_2 or I_3^- , instead of I^- . Given the exceptional adsorption capability of I_2 alone in aqueous solution by both **SUPE-py-Imine-Cage** and **SUPE-py-Amine-Cage** and the negligible pH effect on the adsorption of I_3^- (as I_2/KI), we can conjecture that I_2 adsorption could dominate the apparent I_3^- (as I_2/KI) uptake. In order to understand the details of the iodine capture by the superphane cages, gas-phase molecular dynamics simulation was carried out. As a result, the randomly distributed I_2 molecules were found able to aggregate around the superphane cages over time (Supplementary Figs. 103-104). At the equilibrated state (500 ps), one I_2 molecule was observed to be trapped right within the

superphane cages through hydrogen bonding while other iodine molecules were either adsorbed on the surface of the sorbents (through charge-transfer interactions) or stay in close contact through Van der Waals forces (Supplementary Figs. 103-105). This allows us to suggest that the highly efficient iodine uptake could result from three types of iodine trap, viz. inner binding, surface adsorption and Van der Waals trap.

Supplementary Fig. 69. The removal efficiency for I_2 , I_3^- , I^- and IO_3^- in aqueous solutions at (a) $pH=3$, (b) $pH=7$, (c) $pH=10$ using **SUPE-py-Imine-Cage**; (d) $pH=3$, (e) $pH=7$ and (f) $pH=10$ using **SUPE-py-Amine-Cage**.

Supplementary Fig. 103. Snapshots of molecular dynamics run on the superphane **SUPE-py-Imine-Cage** and I_2 in the gas phase using periodic boundary conditions at 298.15 K at (a) 0 ps, (b) 100 ps and (c) 500 ps. Note: I_2 : **SUPE-py-Imine-Cage** = 50 : 1.

Supplementary Fig. 104. Snapshots of molecular dynamics run on the superphane *SUPE-py-Amine-Cage* and I₂ in the gas phase using periodic boundary conditions at 298.15 K at (a) 0 ps, (b) 100 ps and (c) 500 ps. Note: I₂ : *SUPE-py-Amine-Cage* = 42 : 1.

2. In Supplementary Fig. 7, the characteristic peaks attributed to imine-N and pyridine-N ought to be divided before and after iodine adsorption.

Reply: As suggested, we reworked on the XPS spectra in supplementary Fig. 7 and supplementary Fig. 40 to make the characteristic peaks attributed to imine-N and pyridine-N in XPS spectra divided before and after iodine adsorption.

Supplementary Fig. 7 XPS spectra of N 1s for (a) *SUPE-py-Imine-Cage* and (b) *SUPE-py-Amine-Cage* before and after exposure to iodine vapor for 36 h.

Supplementary Fig. 40 XPS N 1s spectra of (a) *SUPE-py-Imine-Cage* and (b) *SUPE-py-Amine-Cage* before and after adsorption of iodine (I₂/KI).

3. In Supplementary Fig. 8, the annotations corresponding to non-I₂-loaded and I₂-loaded should be provided in figures.

Reply: We thank this reviewer for his/her constructive suggestion. As suggested, the annotations corresponding to non-I₂-loaded and I₂-loaded in Supplementary Fig. 8 are provided in this revised version.

Supplementary Fig. 8 XPS spectra of O 1s for (a) **SUPE-py-Imine-Cage** and (b) **SUPE-py-Amine-Cage** before and after exposure to iodine vapor for 36 h.

4. In page 17, they deemed that the I₃⁻ are combined by hydrogen bonding and charge-transfer interactions between I and N. However, especially for the hydrogen bonding, it can't be supported by current evidence in ¹H NMR spectra with low resolution. Crystal structure of I₃⁻@SUPE-py-Imine-Cage or I₃⁻@SUPE-py-Amine-Cage or use of MD simulations and ab initio molecular dynamics could support their mechanism.

Reply: We thank this referee for his/her great comment and suggestion. As we mentioned before in this letter and in the main text, I₃⁻ is special due to its complex properties, inter alia its dynamic equilibrium $I_2 + I^- \rightleftharpoons I_3^-$. According to the iodine adsorption experiments, I₂ in aqueous solution can be efficiently adsorbed by both superphane cages. In contrast, they failed to uptake anionic I⁻ in water solution even if the adsorption time was extended to 3 h (Supplementary Fig. 69). As such, the actual uptake of the complex I₃⁻ species could be processed in the form of I₂ or I₃⁻, instead of I⁻. Given the exceptional adsorption capability of I₂ in aqueous solution by both **SUPE-py-Imine-Cage** and **SUPE-py-Amine-Cage** and the negligible pH effect on the adsorption of I₃⁻ (as I₂/KI), we can conjecture that I₂ adsorption could dominate the apparent I₃⁻ (as I₂/KI) uptake. In order to understand the details of the iodine capture by the superphane cages, great efforts were devoted to growing single crystals of iodine and the superphane cages, but to no avail. Alternatively, gas-phase molecular dynamics simulation was carried out. As a result, the randomly distributed I₂ molecules were found able to aggregate around the superphane cages (Supplementary Figs. 103-104). At the equilibrated state, one I₂ molecule was observed to be trapped right within the superphane cages through hydrogen bonding (Supplementary Figs. 105).

Supplementary Fig. 105. Snapshots of molecular dynamics run (at 500 ps) on the superphane (a) *SUPE-py-Imine-Cage* and (b) *SUPE-py-Amine-Cage* with iodine in the gas phase using periodic boundary conditions at 298.15 K. These results indicated the occurrence of hydrogen bonding between the trapped iodine and the cages.

5. Adsorption isotherm model for static I_3^- adsorption in aqueous media are supposed to be provided. Besides, the conditions of batch adsorption experiments (e.g., initial concentration of I_3^- and competing anions) should be addressed in caption for Fig 3 and Fig 4.

Reply: As suggested, we provided the static adsorption isotherm for static I_2 and I_3^- adsorption using *SUPE-py-Imine-Cage* or *SUPE-py-Amine-Cage* (along with I_2 and I_3^- adsorption with activated carbons) in aqueous media in this revised version (Supplementary Fig. 46). Meanwhile, the conditions of batch adsorption experiments (e.g., initial concentration of I_3^- and competing anions) were provided in the caption for Fig 3 and Fig 4.

Supplementary Fig. 46. Adsorption isotherm for static adsorption of (a) I_2 and (b) I_3^- (I_2/KI) in aqueous media with *SUPE-py-Imine-Cage*, *SUPE-py-Amine-Cage* and activated carbons (ACs).

6. For dynamic adsorption, the breakthrough curve should be plotted and corresponding analysis should be provided.

Reply: Again, we do thank this reviewer for his constructive suggestion. We have plotted the breakthrough curve for *SUPE-py-Imine-Cage*, *SUPE-py-Amine-Cage* and ACs, respectively. Corresponding analysis was provided in the manuscript as follows:

*“The breakthrough curves were then obtained from the adsorption of aqueous I_2 and I_3^- on *SUPE-**

py-Imine-Cage, *SUPE-py-Amine-Cage* or ACs in a fixed-bed column. Herein, we defined the value of C/C_0 at 0.05 as the breakthrough point, where C_0 is the initial concentration of sorbate (mg/L), C is the desired concentration of sorbate at time t (mg/L). At the breakthrough point, 95% removal efficiency for iodine in water was achieved. As a result, the breakthrough volume of **SUPE-py-Imine-Cage** for iodine removal was estimated to be 60 mL for I_2 and 3 mL for I_3^- , respectively. Notably, the latter smaller breakthrough volume could be attributed to the much higher initial concentration of I_3^- than that of I_2 (3000 mg/L VS ~300 mg/L). Similarly, the breakthrough volume of **SUPE-py-Amine-Cage** was tested to be 40 mL for I_2 and 1.5 mL for I_3^- , separately (Supplementary Fig.84). In contrast, the breakthrough volume for ACs was estimated to be 6 mL for I_2 and 1 mL for I_3^- , respectively, under the same conditions (Supplementary Fig. 84). In aggregate, these findings allowed us to suggest that **SUPE-py-Imine-Cage** and **SUPE-py-Amine-Cage** have higher adsorption affinity toward I_2 and I_3^- in aqueous solution than commercially available ACs."

Supplementary Fig. 84 Breakthrough curve of the ACs, **SUPE-py-Imine-Cage** and **SUPE-py-Amine-Cage** for iodine removal from (a) saturated I_2 aqueous solution and (b) I_3^- aqueous solution (600 mg KI and 300 mg I_2 in 100 mL of H_2O).

7. To figure out the contribution for ionic I^- and molecule I_2 , the supplementary adsorption experiments in solution for single I^- (in water) and single I_2 (in hexane) are supposed to be conducted. **Reply:** This is a great point. As suggested, we conducted additional adsorption experiments in solution for single I^- (5 mM in water) and single I_2 (5 mM in hexane). As a result, in *n*-hexane, both cages, viz. **SUPE-py-Imine-Cage** and **SUPE-py-Amine-Cage**, were found able to remove ca. 99 % of I_2 in 8 h and 3 h, respectively (Supplementary Fig. 70). Nevertheless, in water, both **SUPE-py-Imine-Cage** and **SUPE-py-Amine-Cage** were not able to adsorb I^- at all even if the adsorption time was extended over 8 h or 3 h. This led us to conclude that both cages are mainly able to efficiently adsorb I_2 molecules instead of single I^- .

Supplementary Fig. 70 I_2 (in *n*-hexane) and I^- (in water) removal efficiency of (a) *SUPE-py-Imine-Cage* and (b) *SUPE-py-Amine-Cage*, respectively.

Response to the comments of Reviewer #2:

Comments: The manuscript entitled “Nonporous amorphous superadsorbents for highly effective and selective adsorption of iodine in water” submitted to be considered for publication in *Nat. Commun.* describes the synthesis and characterization of a hexa-imine superphane dynamic covalent cage and its covalent amine counterpart. They evaluate their use in I_2 and I_3^- adsorption from I_2 vapor and from aqueous sources either in batch or in flow. They claim that the prepared superphane cages have superior adsorption properties than any of the previously porous adsorbents reported to date.

The work is very similar in terms of experimental procedures and discussion to the one reported last year in *Angew. Chem. Int. Ed.* (ref 18 in the original manuscript) from Ji, Sessler, Wang and collaborators. The main differences with the previously reported work are basically the receptor used as adsorbent (superphane vs calix[4]pyrrole-based crosslinked polymer) and the superior uptake capacity described in the present work (iodine vapor adsorption capacity 9 g g^{-1} vs 3 g g^{-1} , respectively, and iodine capture from water 5 g g^{-1} vs 3 g g^{-1} , respectively). The differences in uptake capacity are quite significant. However, I would suggest additional experiments to confirm these results.

Reply: *We thank for the reviewer’s positive comments, and we have conducted additional experiments to confirm these results as suggested by this reviewer (see below).*

In the present work the authors only report the uptake capacities, kinetics, and recovery with the imine and amine cages prepared. They compare their results with the reported ones but they do not report any control experiment performed exactly in the same conditions they use. I strongly suggest to perform the same experiments using a typical activated carbon and compare their results under the same experimental conditions and applying the same data analysis. Having their own control experiment will discard experimental differences on the obtained data compared to described data and will provide more reliable conclusions.

Reply: *We do appreciate this reviewer’s insightful suggestion. As suggested, we performed the control experiment using commercially available activated carbon under the same conditions we use. As a result, both superphane based cages proved much more efficient and selective for iodine*

uptake in aqueous media than commercially available activated carbon. For more details, please refer to the revised manuscript and ESI.

The work is written in a clear manner and most of the experimental procedures are well-described. However, I suggest to clarify how did they control the equivalents of I₂ added in the NMR titration data included in the SI (S10 and S12).

Reply: *Again, many thanks to this reviewer's positive comment and we are really sorry for, inadvertently, missing these details. In this revised version, we added these details in question into the ESI (e.g. original Supplementary Figs. S10 and S12). The equivalents of I₂ added in the NMR titration data was controlled by continuously adding aliquots of I₂ in CDCl₃ (200 mM) into a solution of **SUPE-py-Imine-Cage** (2.0 mM, 500 μL) or **SUPE-py-Amine-Cage** (2.0 mM, 500 μL) using a microsampler.*

Additionally, I recommend the authors to address the following issues in the revised version of the manuscript:

1. The authors use ¹H NMR spectroscopy to support the binding of iodine with imine and amine cages. Figures S9-S12 in the supporting information show different exchange kinetics in the ¹H chemical shift time scale for imine and amine cages (fast for imine and fast for amine). The authors do not mention anything about this in the main text. I suggest to add a comment on that to explain the differences observed. Is the binding mechanism somehow related to the adsorption mechanism? Please add a comment on that.

Reply: *This is a great point. As suggested, we have added a comment on the ¹H NMR spectroscopy to support the binding of iodine with **SUPE-py-Imine-Cage** and **SUPE-py-Amine-Cage** in this revised version as follows:*

*“Notably, the binding of I₂ with **SUPE-py-Imine-Cage** displayed fast exchange kinetics on the ¹H NMR time scale, in contrary to the slow exchange kinetics of I₂ binding with **SUPE-py-Amine-Cage**, suggesting that the reduced superphane cage is likely to bind iodine in a more favorable manner presumably due to the occurrence of multiple secondary amine units as extra binding sites.”*

*In our opinion, study of the binding mechanism at molecular level could somewhat advance our understanding of the interactions between the adsorbent and the iodine, as well as the adsorption mechanism. However, in our current system, we found that the binding mechanism doesn't dominate the adsorption mechanism as inferred from the fact that **SUPE-py-Imine-Cage** always displayed higher iodine adsorption capability than **SUPE-py-Amine-Cage** and, in the sorbent recycling experiment induced by temperature and vacuum, I₂@**SUPE-py-Amine-Cage** is likely to desorb more I₂ than I₂@**SUPE-py-Imine-Cage**. This inconsistency with the binding mechanism could be rationalized by the fact that **SUPE-py-Imine-Cage** has more groups (including imine and pyridinyl units) to be polarized, generating more favorable charge transfer interactions than **SUPE-py-Amine-Cage**. Meanwhile, charge-transfer interactions between the nitrogen-bearing groups (imine and pyridinyl units) and iodine tend to be favored in solid state.*

2. The authors only report the adsorption kinetics for the uptake in vapor phase. What about the studies in aqueous solution? I suggest to add also these values and compared them with a control

adsorbent (activated carbon).

Reply: We are grateful for this reviewer's constructive suggestion. As we mentioned above, in this revised version, we have supplemented the adsorption kinetics for I_2 and I_3^- uptake in aqueous solution using **SUPE-py-Amine-Cage**, and **SUPE-py-Imine-Cage**, as well as activated carbon as the control adsorbent (Supplementary Fig. 46).

Supplementary Fig. 46. Adsorption isotherm for static adsorption of (a) I_2 and (b) I_3^- in aqueous media with **SUPE-py-Imine-Cage**, **SUPE-py-Amine-Cage** and activated carbons (ACs).

3. The authors claim that they can recycle the adsorption material by sonication in isopropanol. I wonder if they could simply use temperature and vacuum to recycle the material as done for previously reported materials. Please add a comment on that.

Reply: Nice question. We conducted the experiments, using temperature and vacuum, to recycle the adsorbents as done for previously reported materials. Specifically, 10 mg of **SUPE-py-Imine-Cage** or **SUPE-py-Amine-Cage** was subjected to adsorb gaseous iodine until adsorption saturation was nearly reached (5.80 g g^{-1} for **SUPE-py-Imine-Cage** or 4.47 g g^{-1} for **SUPE-py-Amine-Cage**). The mixtures were then subjected to desorption at 75°C or under vacuum. As a result, the adsorbed iodine was gradually released from the sorbents within 6 h at 75°C and ambient pressure (Supplementary Fig. 91). In contrast, I_2 was observed to desorb from the sorbents sharply at 75°C and under vacuum within 1 h, reaching plateaus at (3.49 g g^{-1} for **SUPE-py-Imine-Cage** or 2.24 g g^{-1} for **SUPE-py-Amine-Cage**) within 4 h. This observation indicated that, alternatively, superphane cages can be recycled by simply using temperature and vacuum.

Supplementary Fig. 91 The iodine desorption profile of $I_2@$ **SUPE-py-Imine-Cage** (in black) and $I_2@$ **SUPE-py-Amine-Cage** (in red) (a) at 75°C under ambient pressure and (b) at 75°C under vacuum.

4. The authors show the iodine adsorption at different pHs 3-10. They refer always to I_3^- and I_2 extraction. At different pHs in aqueous media other forms of dissolved I are possible: I^- , IO_3^- , HOI, etc. I suggest to add a comment on how this can affect the selectivity and the adsorption capacity of the cages. Please add a comment on which as the basis of the observed selectivity for I_2 and I_3^- over the interfering anions tested.

Reply: Again, we do appreciate this reviewer's great and professional comments. Because of the unstable nature of hypoiodous acid, HOI was unavailable for adsorption test. As such, we carried out extra adsorption experiments with I^- and IO_3^- (K^+ as the counter cations), under the same conditions of I_2 and I_3^- (K^+ as the counter cation) adsorption experiments. As a result, both superphane cages, viz. **SUPE-py-Imine-Cage**, and **SUPE-py-Amine-Cage**, were found unable to uptake neither I^- nor IO_3^- in aqueous solutions at different pHs (pH = 3, 7 and 10), in marked contrast to the exceptionally high iodine adsorption capability and selectivity (Supplementary Fig. 69). So, over the course of iodine adsorption at different pHs 3-10, in spite of other forms of dissolved iodine species, e.g. I^- and IO_3^- , I_3^- and I_2 extraction could dominate the iodine uptake capability and selectivity. We updated these findings in the main text and ESI.

Supplementary Fig. 69 The removal efficiency of I_2 , I_3^- , I^- and IO_3^- in aqueous solutions at (a) pH=3, (b) pH=7, (c) pH=10 using **SUPE-py-Imine-Cage** and in aqueous solutions at (d) pH=3, (e) pH=7 and (f) pH=10 using **SUPE-py-Amine-Cage**.

Response to the comments of Reviewer #3:

Comments: The field of non-porous amorphous materials (yet capable of adsorption) is very new and recent with few examples. Using a simple covalent cage is attractive and opens a new paradigm because the simplicity of the approach suggests that many previously synthesized cages (overlooked) and more to come are capable of similar things and perhaps (actually probably) not only restrained to I_2 adsorption. The simplicity of avoiding ordering the matter in 3D to get a crystalline material is of huge interest notably for industry. Rest the evaluation of energetic and product cost of production and re-activation which should be mentioned as perspectives but falls beyond the scope of this fundamental research in my opinion. The exceptional performances in terms of iodine capture in harsh (pH) and competitive (anions) conditions plus the highest capacities so far from aqueous

solutions are sufficient arguments, to me, to justify publication for a leadership journal of broad audience. However, I have several concerns which I believe should be addressed before publication.

Reply: *We are really grateful for this reviewer positive comments. We did our best to address this reviewer's concerns as below.*

1. Introduction: Please explain better in which context these 2 radionuclides ^{129}I and ^{131}I are produced. Is it during normal civil nuclear energy exploitation or after a nuclear accident? (both?) Otherwise we do not understand why this is urgent to develop such materials to catch dangerous isotopes of iodine. To me it is not a pressant present need but it could become very urgently needed in case of problems. So I appreciate the initiative to anticipate and possibly make available solutions if dramatic issues raise in the future.

Reply: *Great point. It is known that ^{129}I and ^{131}I , along with other radioactive species, are mainly produced from nuclear (plutonium-239 and uranium-235) fission over the course of generating nuclear power. The resulting urgent safety issue is related to the administration of nuclear waste and nuclear accidents. For instance, the Chernobyl (in 1986) and Fukushima (in 2011) nuclear disasters both led to the release of large quantities of radioactive iodine, including ^{129}I and ^{131}I , into the atmosphere and water bodies, posing sudden threat to people's safety and health. Moreover, illegal/improper disposal of radioactive wastes (including ^{129}I and ^{131}I) could lead to release of large quantities of radioactive iodine into the ecosystems and eventually into the water bodies. As such, it is urgently needed to develop advanced materials, like our current systems, to capture iodine from mixture gas and, inter alia, aqueous solutions with fast kinetics and high adsorption capacity at room temperature to possibly make available solutions if dramatic issues raise in the future. We reshaped the introduction part in this revised version to make it more readable and understandable.*

2. Explain that 1st gaseous I_2 needs to be trapped by specific sorbents and then 2nd dissolved I_2 in water which is the focus of this paper.

Reply: *Again, we thank this reviewer for his/her constructive suggestion. As mentioned above, radioactive iodine, a product of plutonium-239 and uranium-235 fission, is volatilized (e.g. as I_2) into the atmosphere during the reprocessing of nuclear fuel, becoming an important environmental concern. Meanwhile, water cooling of nuclear fission reactors can result in a direct aqueous contaminant radioactive iodine and nuclear accidents such as the Chernobyl and Fukushima nuclear disasters, could lead to the direct release of large quantities of radioactive iodine into water bodies. Moreover, radioactive isotopes are widely used for diagnostic and therapeutic applications, resulting in an ancillary radioactive iodine source.*

3. When citing NACs, actually non-porous adaptive crystals are much older: Science Atwood, 2002 (Storage of Methane and Freon by Interstitial van der Waals Confinement). Please state this clearly. Other papers appeared after that and before NACs like Gas-induced transformation and expansion of a non-porous organic solid, Nat Mater 2008 or Diffusion of vaporous guests into a seemingly non-porous organic crystal, ChemComm 2014.

Reply: *As suggested, we cited these early works mentioned by this reviewer in this revised version. Thanks.*

4. Page 2, "unwanted interference of water molecules", this critical point is justly raised by the

authors. I would just add this paper in the proposed reference list as a noticeable example/exception: “Angew. Chemt. Int Ed. Energy-efficient iodine uptake by a molecular host guest crystal, 2022”.

Reply: Nice suggestion. We cited this paper mentioned by this reviewer as a noticeable exception in this revised version.

5. Why perform I₂ uptake at 348K? In water or atmosphere, the temperature is likely not 348K. Please justify both in the letter and in the manuscript for readers and clarity.

Reply: This is a great point. Like other reported systems in the literatures for gaseous I₂ uptake at 348 K, we perform I₂ vapor uptake at 348 K as well, which is based on the typical nuclear fuel reprocessing condition of nuclear industry. We clarify this in the main text in this revised version. However, in aqueous solutions, iodine adsorption experiments were carried out at room temperature instead of 348 K.

6. Figure 2d and corresponding text (and later Figure 5d): there is a problem. Data suggest formation of anionic iodine species I₃⁻ and I₅⁻. However, this leaves the charge balance unequilibrated and positive species must have formed but there is no data or assumption of what happened there. Please explain or suggest a mechanistic explanation of what happened (transformation of I₂ into I₃⁻ or I₅⁻ and which cation formed to balance the charge to 0).

Reply: This is a great question. According to the reported adsorbent materials doped with multiple N sites (e.g. pyridinyl and imine units) for I₂ adsorption, the mechanical studies revealed that the occurrence of charge or electron transfer from the N-containing sites to I₂ is usually observed, resulting in the reversible charge-transfer products via N⁺⋯I₃⁻ or N⁺⋯I₅⁻ interactions (Chem. Mater. 2018, 30, 2299–2308, J. Am. Chem. Soc. 2013, 135, 16256–16259., and J. Am. Chem. Soc. 2011, 133, 14920–14923.). Analogous to what has been seen in many reported systems, our current superphane cages consisting of 6 pyridinyl units and 12 imine/amine groups are able to form stable charge-transfer species with I₂, which could be fully supported by XPS results (Figure 2d and Supplementary Fig. 7). As such, in the case of the I₂ adsorption, the counter positive species of anionic iodine I₃⁻ and I₅⁻ would be cationic ammonium, enabling the charge to be balanced to 0.

Supplementary Fig. 7 XPS spectra of N 1s for (a) SUPE-py-Imine-Cage and (b) SUPE-py-Amine-Cage before and after exposure to iodine vapor for 36 h.

Supplementary Fig. 41 XPS N 1s spectra of (a) *SUPE-py-Imine-Cage* and (b) *SUPE-py-Amine-Cage* before and after adsorption of I₃⁻ (I₂/KI) in aqueous media.

7. What is the reason for the specificity for I₃⁻ anions versus other anions? Please suggest a rationale.

Reply: As the reviewer may appreciate, I₃⁻ is quite special and complex than other anions due in large part to its highly dynamic equilibrium $I_2 + I^- \rightleftharpoons I_3^-$. The adsorption/removal of I₃⁻ doesn't mean the direct uptake of iodine in the form of I₃⁻. Instead, it could be removed via any species of I⁻, I₂, and I₃⁻, especially the latter two. As mentioned above, both superphane cages were found unable to adsorb single I⁻ from aqueous solutions. But as we stated in this work, both superphane cages can serve as highly efficient and selective adsorbents for uptake of I₂ in aqueous media. As such, I₂ adsorption probably dominates the uptake capability and specificity for I₃⁻ versus other anion. Direct I₃⁻ uptake by the cages are possible to contribute to the exceptional iodine adsorption capability.

8. It lacks a ratio between the number of cage molecules and the number of trapped I₂ and I₃⁻. For example, giving the correct numbers in the form X molecules of I₂ per cage molecule would help the reader to picture how many molecules of interest have been trapped by 1 molecule of cage. This would also help decipher the location of trapped molecules for example if relatively high, this would be an additional argument toward intermolecular association if intramolecular is privileged. This should be briefly discussed.

Reply: This is an interesting yet challenging question. As is well established in the literatures, I₂ was usually captured in the form of I₂, I₃⁻, and I₅⁻ by rich nitrogen-bearing materials with the occurrence of charge transfer from nitrogen atoms to iodine. So it is most difficult to determine the exact numbers of each iodine species captured within the sorbents. But we can rationally treat the adsorbed iodine, regardless of its specific forms, as the starting adsorbate, e.g. I₂, from which all iodine species are originated. Along this line, in the gas phase, the superphane cages were estimated to apparently capture 36 I₂ per *SUPE-py-Amine-Cage* and 28 I₂ per *SUPE-py-Imine-Cage*, respectively. In aqueous solutions, one *SUPE-py-Amine-Cage* molecule was estimated to trap up to 50 I₂ while one *SUPE-py-Imine-Cage* molecule was approximated to capture 42 I₂. Obviously, the cages are not possible to encapsulate so many I₂ inside the cavities of the cages. Thus most I₂ are expected to be trapped between cages.

9. About the very few previous examples of non-porous amorphous materials yet capable of adsorption, I advise the authors to add these references which are clearly in the scope of this paper:

Adsorptive Separation of Benzene, Cyclohexene, and Cyclohexane by Amorphous Nonporous Amide Naphthotube Solids, *Angew Chem Int Ed* 2020, 59, 19945-19950; *J. Phys. Chem. C* 2011, 115, 47, 23344; *Bulletin of Chemical Reaction Engineering & Catalysis*, 12 (2), 263-272 (doi:10.9767/bcrec.12.2.766.263-272; the following paper is advised to be cited for readers: *CrystEngComm*, 2012,14, 1909-1919. I also invite the researchers to very carefully read this paper (DOI: 10.1002/adfm.200800624) which should merit high attention from the community stating that measuring a N₂ BET alone may not be sufficient to state about the absence of porosity in a material. I invite the authors to keep their statements about nonporosity but add a sentence saying something that nevertheless, previous examples have shown that BET alone may not be enough to fully state the absence of porosity in a solid and that other experiments studies are needed to fully ascertain this point. This in my opinion does not impede the relevance of the work nor decreases the quality of the material which was proven on multiple occasions to be fully relevant for iodine iodide capture in (much appreciably) efforts toward more real conditions of use).

Reply: We do appreciate this reviewer's professional and constructive comment. According to this reviewer's suggestion, we cited these references and made proper discussion accordingly in this revised version.

10. Why tBu groups are present in the superphane cages? Please explain briefly why it has been introduced in the syntheses.

Reply: Honestly, we initially attempted to synthesize another superphane cage without tBu groups, but to no avail. This was attributed to the poor solubility of the hexakis-aldehyde precursor and the superphane cages, impeding the purification and characterization. The bulky and aliphatic tBu groups were introduced to improve the solubility of the key precursors and desired superphane cages. A comment was made in the main text accordingly.

11. Page S5, crude formula of compound 6 is incorrect, I found C₇₈H₉₀O₁₂N₁₂ and so the MS. Please correct.

Reply: We thank this reviewer's good eye. We made the correction accordingly in the ESI.

12. Supplementary Figures containing XPS spectra, please explain how the signal deconvolution was done. It can be by a simple sentence or paragraph in the general information, early in ESI.

Reply: As suggested, we have added a paragraph to describe the details of the XPS signal deconvolution in the general information, early in ESI as follows:

"Deconvolution was usually applied to process low energy-resolution data for possible comparison with data collected at high resolution. The raw XPS spectral data were firstly baseline-corrected and were analyzed using a peak-fitting method. This involved assigning initial peak positions based on expected binding energy values for the elements of interest, along with expected peak widths based on typical chemical shifts compared with the reported data in literature. The fitting process was iterative, that involved adjusting the peak parameters (such as position, width, and shape) until the best fit to the original data was achieved, as determined by a least-squares fitting algorithm."

13. Tests with anions, no salt in indicated so we do not know which cation was used, please indicate. Also explain (in the paper) the rationale for the use of these anions.

Reply: Sorry for missing such details. As mentioned in the main text, I_3^- anion (as its K^+ salt) was in-situ obtained by mixing KI and I_2 in water via dynamic equilibrium $I_2 + KI \rightleftharpoons KI_3$. Likewise, other tested anions were also introduced as their potassium (K^+) salts. As to the anions used, according to the literatures, Cl^- , Br^- , NO_3^- and SO_4^{2-} are the major anionic species in the real-world nuclear wastes so that they are commonly used as the competitive anions to evaluate the selectivity of the adsorbents of interest toward iodine.

14. Stability experiments of cages exposed to different pH are very nice. Comparisons with known compounds, very nice and insightful.

Reply: Many thanks to this reviewer for his/her positive comment.

15. Something remains unclear. Why after having trapped X equiv. of I_2 or I_3^- , these species continue to be attracted to reach such high levels of adsorption? We could rationally expect that, after having trapped a few equiv. of iodine species, the process would stop? Please try to propose a hypothesis to account for this result (internal and external adsorption, I_2 - I_2 interactions or I_3^-/I_3^- interactions, hydrophobic effect, etc ... which would make sense to the authors.

Reply: The central cavities (in pink) of both superphane cages are well-defined and finite to accommodate only one I_2 molecule inside each cage (Supplementary Fig. 106). However, there exist six sub-cavities surrounding each central cavity. We could imagine that the empty cage might be in a flat conformation without the occurrence of pores as solids. The iodine binding in the central cavities could lead to the "swelling" of the entire cages to expand its outer apertures, allowing for further binding of iodine either in the outer cavities or between the cages. This was reminiscent of a balloon-blowing process. At the beginning, the balloons are flat with little space inside and outside the balloons. However, the balloon blowing not only results in the occurrence of inner pores filled with air but also offers large amount free space between the inflated balloon (Supplementary Fig. 101). Thus, the high iodine adsorption capability of both superphane cages could result from internal and, inter alia, external adsorption of iodine through multiple intermolecular interactions, presumably including hydrogen bonding, charge-transfer interactions, I_2 - I_2 interactions and I_2/I_3^- interactions.

Supplementary Fig. 106 Possible iodine binding domains of the central cavities (in pink) and outer apertures (in green) for (a) *SUPE-py-Imine-Cage* and (b) *SUPE-py-Amine-Cage*.

16. Something not so surprising is the adsorption of I_3^- from water as, iodine in the form of I_3^- is adsorbed when cages are exposed to I_2 . Nevertheless, why did the authors studied I_2 and I_3^- but not

I⁻? They mention formation of I₃⁻ from I₂ and I⁻ so, to form I₃⁻, I⁻ must be present. Please provide results with I⁻, even if not good, results are so excellent with the two other species (I₂ and I₃⁻) that it deserves publication.

Reply: This is a great question. As suggested, we tested adsorption of I⁻ from water using the *SUPE-py-Imine-Cage* and *SUPE-py-Amine-Cage* under the same conditions. As a result, no or negligible I⁻ adsorption was observed using neither *SUPE-py-Imine-Cage* nor *SUPE-py-Amine-Cage*, even if the adsorption time was extended to 3 h at pHs 3-10 as monitored by ion chromatography (Supplementary Fig. 69). This observation was in sharp contrast to the fast kinetics and high capability of I₂ or I₃⁻ uptake using either *SUPE-py-Imine-Cage* or *SUPE-py-Amine-Cage*.

Supplementary Fig. 69 The removal efficiency of I₂, I₃⁻, I⁻ and IO₃⁻ in aqueous solutions at (a) pH=3, (b) pH=7, (c) pH=10 using *SUPE-py-Imine-Cage* and in aqueous solutions at (d) pH=3, (e) pH=7 and (f) pH=10 using *SUPE-py-Amine-Cage*.

17. Instead of cavity self-filling I would talk about void or cavity collapse upon material activation, if so.

Reply: Again, we are grateful for this reviewer's insightful suggestion. We talked about void or cavity collapse upon material activation in this revised version.

18. CCDC numbers are not given for the crystal structures.

Reply: As suggested, we added the CCDC numbers (2245537 for *SUPE-py-Imine-Cage* and 2245538 for *SUPE-py-Amine-Cage*) both in the main text and ESI.

REVIEWERS' COMMENTS

Reviewer #1 (Remarks to the Author):

I have no further comments. This work can be published in its current form.

Reviewer #2 (Remarks to the Author):

I'm happy to confirm that the authors have addressed most of the comments of the reviewers correctly. I would recommend the acceptance of the manuscript for publication in Nature Communications after addressing the minor issues stated below. No further revisions are needed from this reviewer.

1) I suggest the authors to rewrite the clarification related to the I₂ number of equivalents. "1H NMR titrations of SUPE-py-Imine-Cage (2.0 mM, 500 µL) and SUPE-py-Amine-Cage (2.0 mM, 500 µL) with I₂ were carried out by adding small aliquots of a solution of I₂ in CDCl₃ (200 mM) into a solution of the host in the same solvent using a microsampler."

2) As suggested by the reviewer, the authors identified the different exchange kinetics in the 1H NMR timescale observed for the binding of iodine with the two different cages. They attributed the slow exchange kinetics observed for the amine-Cage to a more favorable binding of iodine to this cage compared to the imine analogue. However, the exchange kinetics does not necessarily correlate with the thermodynamics of the process but just to the binding kinetics (different binding mechanisms operating in the different cages). Authors should remove the correlation between exchange kinetics and thermodynamics.

Reviewer #3 (Remarks to the Author):

I think the authors addressed well my comments and so the paper can be published now.

Response to the comments of Reviewer #1:

Reviewer #1 (Remarks to the Author):

I have no further comments. This work can be published in its current form.

Reply: We do thank this reviewer for recommending this paper for publication in Nature Communications.

Response to the comments of Reviewer #2:

Comments: Reviewer #2 (Remarks to the Author):

I'm happy to confirm that the authors have addressed most of the comments of the reviewers correctly. I would recommend the acceptance of the manuscript for publication in Nature Communications after addressing the minor issues stated below. No further revisions are need from this reviewer.

Reply: Again, we are really grateful for this reviewer's constructive comments, and we tried our best to address the minor issues stated below.

1) I suggest the authors to rewrite the clarification related to the I₂ number of equivalents.

"¹H NMR titrations of SUPE-py-Imine-Cage (2.0 mM, 500 μL) and SUPE-py-Amine-Cage (2.0 mM, 500 μL) with I₂ were carried out by adding small aliquots of a solution of I₂ in CDCl₃ (200 mM) into a solution of the host in the same solvent using a microsampler."

Reply: As suggested, we rewrote the clarification related to the I₂ number of equivalents in the SI.

2) As suggested by the reviewer, the authors identified the different exchange kinetics in the ¹H NMR timescale observed for the binding of iodine with the two different cages. They attributed the slow exchange kinetics observed for the amine-Cage to a more favorable binding of iodine to this cage compared to the imine analogue. However, the exchange kinetics does not necessarily correlate with the thermodynamics of the process but just to the binding kinetics (different binding mechanisms operating in the different cages). Authors should remove the correlation between exchange kinetics and thermodynamics.

Reply: We do appreciate this reviewer's professional and constructive comments. As suggested, we removed the correlation between exchange kinetics and thermodynamics. Many thanks.

Response to the comments of Reviewer #3:

Reviewer #3 (Remarks to the Author):

I think the authors addressed well my comments and so the paper can be published now.

Reply: We do thank this reviewer for recommending this paper for publication in Nature Communications.